# WNT5A is transported via lipoprotein particles in the cerebrospinal fluid to regulate hindbrain morphogenesis

Karol Kaiser [1,2], Daniel Gyllborg [2], Jan Procházka[3], Alena Salašová[2,4], Petra Kompaníková[1],
Francisco Lamus Molina[5], Rocio Laguna-Goya[6], Tomasz Radaszkiewicz[1], Jakub Harnoš[1], Michaela Procházková[3],
David Potěšil[7], Roger A. Barker[6], Ángel Gato Casado[5], Zbyněk Zdráhal[7], Radislav Sedláček[3], Ernest Arenas [2],
J. Carlos Villaescusa [1,2,8] & Vítězslav Bryja [1]

WNTs are lipid-modified proteins that control multiple functions in development and disease via short- and long-range signaling. However, it is unclear how these hydrophobic molecules spread over long distances in the mammalian brain. Here we show that WNT5A is produced by the choroid plexus (ChP) of the developing hindbrain, but not the telencephalon, in both mouse and human. Since the ChP produces and secretes the cerebrospinal fluid (CSF), we examine the presence of WNT5A in the CSF and find that it is associated with lipoprotein particles rather than exosomes. Moreover, since the CSF flows along the apical surface of hindbrain progenitors not expressing *Wnt5a*, we examined whether deletion of *Wnt5a* in the ChP controls their function and find that cerebellar morphogenesis is impaired. Our study thus identifies the CSF as a route and lipoprotein particles as a vehicle for long-range transport of biologically active WNT in the central nervous system.

[1] Department of Experimental Biology, Faculty of Science, Masaryk University, Brno 62500, Czech Republic. [2] Division of Molecular Neurobiology, Department of Medical Biochemistry and Biophysics, Karolinska Institutet, Stockholm 171 77, Sweden. [3] Czech Centre for Phenogenomics and Laboratory of Transgenic Models of Diseases, Institute of Molecular Genetics of the CAS, v. v. i., Prumyslova 595, Vestec 252 42, Czech Republic. [4] Danish Research Institute of Translational Neuroscience, Department of Biomedicine, Aarhus University, Aarhus C 8000, Denmark. [5] Departamento de Anatomía y Radiología, Facultad de medicina, Universidad de Valladolid, Ramón y Cajal 5, 47005 Valladolid, Spain. [6] John van Geest Centre for Brain Repair and Cambridge Stem Cell Institute, University of Cambridge, Cambridge CB2 0PY, UK. [7] Central European Institute of Technology, 625 00 Brno, Czech Republic. [8] Psychiatric Stem Cell Group, Neurogenetics Unit, Center for Molecular Medicine, Department of Molecular Medicine and Surgery, Karolinska University Hospital, Stockholm 171 76, Sweden. Correspondence and requests for materials should be addressed to E.A. (email: ernest.arenas@ki.se) or to J.C.V. (email: carlos.villaescusa@gmail.com) or to V.B. (email: bryja@sci.muni.cz)

Wnt proteins (Wnts) are key players in development and in adult organisms[1]. A crucial feature underlying their function is the ability to spread over long distances leading to formation of concentration gradients, which elicit and orchestrate diverse patterning decisions during development[2]. Post-translational modification, addition of lipid moieties in the endoplasmic reticulum by the acyltransferase Porcupine[3,4], plays a central role in the proper maturation of Wnt ligands and their secretion. The presence of essential lipid modifications in fully processed Wnts[5] represents a challenge for unhindered transport of Wnts in the water-based extracellular space. Several mechanisms for long-range transport of Wnts have been proposed in *Drosophila*, such as lipoprotein particles[6], incorporation into exosomes[7–9], and direct binding to the transporter protein Swim[10]. Other proposed modalities include transport of Wnts via specialized filopodia such as cytonemes in *Drosophila*[11] or zebrafish[12], or via migrating cells, such as neural crest cells in chicken[13]. In mammals, Wnts have been found to be transported via exosomes in the epididymal fluid of mice[14] and in vitro, via binding to the transport protein Afamin[15]. However, the relevance of some of these mechanisms to the mammalian physiology remains unclear[16].

A previous study has reported that deletion of *Otx2* in the hindbrain choroid plexus (HbChP) increases *Wnt4* expression in this structure as well as the levels of WNT4 in the cerebrospinal fluid (CSF)[17]. This study proposed a role of the ChP in regulation of WNT4 secretion into the CSF and WNT signaling at a distant site. However, it is unclear whether WNT4 does really control proliferation in a direct manner at a site distant to where it is produced in vivo. Moreover, it remains to be determined how can a lipophilic molecule such as WNT be transported via the CSF. We thus decided to examine the capacity of the ChP to secrete WNT proteins into the CSF and investigated the mechanism of transport of WNT proteins in the CSF.

Our results show that the embryonic HbChP, but not the telencephalic choroid plexus (TelChP), specifically expresses and secretes high levels of WNT5A into the CSF. Mechanistically, our data indicate that WNT5A preferentially associates to lipoprotein particles, rather than exosomes. Moreover, analysis of hindbrain progenitors that do not express *Wnt5a* and do not have access to WNT5A protein from neighboring cells revealed a morphogenetic defect upon *Wnt5a* deletion. Thus, our result identifies WNT5A as a key regulator of morphogenic behavior of dorsal hindbrain progenitors near, but not adjacent, to the ChP and identify lipoprotein particles as the mechanism of transport of biologically active WNT proteins in the CSF.

## Results

**Distinct expression of *Wnt5a* in the choroid plexuses.** To identify the Wnt family members expressed in the various ChPs, we first analysed expression profiles of all Wnt ligands by means of in situ hybridization at mouse embryonic day (E) 13.5 (Supplementary Fig. 1a). *Wnt5a* was the *Wnt* with the strongest expression in the HbChP (located in the fourth ventricle) (Fig. 1a, b and Supplementary Fig. 1a). *Wnt5a* expression was maintained from E12.5 to E17.5, as assessed by qPCR (Fig. 1c). These results were further corroborated by in situ analysis at E13.5 and E17.5 (Fig. 1d). Notably, *Wnt5a* was not detected in the TelChP (located in the lateral ventricle) (Fig. 1b, c), a result in line with previous findings[18]. On the other hand, *Wnt5a* was found in the adjacent cortical hem (CH)[19] (Fig. 1b and Supplementary Fig. 1a, asterisks), where *Wnt2b, 3a, 7a, 7b, 8b,* and *9a* are also expressed (Supplementary Fig. 1b). Interestingly, high *Wnt5a* expression was found in the epithelium of the HbChP (Fig. 1b, d and Supplementary Fig. 2a), while *Wnt5b* expression was very low at

E13.5 (Supplementary Fig. 1a) or nearly undetectable at E14.5 (Supplementary Fig. 2b–d), suggesting that a possible redundancy between these two Wnts is unlikely.

Using a specific WNT5A antibody, validated in the *Wnt5a-/-* (*Wnt5a^KO*) mice (Supplementary Fig. 3a, b), we found that WNT5A protein localizes to the HbChP from E12.5 to E17.5, but it is absent in the TelChP (Fig. 1e, f). At later embryonic and early postnatal stages, the expression and protein levels of WNT5A in the HbChP progressively decreased (Fig. 1d and Supplementary Fig. 3c, d), suggesting a role of WNT5A in the HbChP during embryonic development. WNT5A was typically found in the apical part of the cytoplasm of secretory epithelial cells, and on occasion, in punctuate structures close to or above the apical cell membrane stained with Aquaporin-1 (AQP1)[20] (Fig. 1g).

Analysis of human fetal brains (week 9 post conception) confirmed that WNT5A protein is found in the HbChP, but not in the TelChP, also in humans (Fig. 2a). Moreover, WNT5A was also identified in the apical part of the cytoplasm of the HbChP epithelium (Fig. 2b) and in punctate structures close to or above the AQP1+ apical membrane of epithelial cells, in direct contact with CSF (Fig. 2c, arrowheads), a result suggestive of WNT5A being secreted from the HbChP to the CSF.

We next examined whether the HbChP expresses mouse *Wntless* (*Wls*; also known as *Gpr177*), a gene encoding a protein indispensable for WNT secretion[21,22]. qPCR analysis revealed that *Wls* is expressed in a similar pattern as *Wnt5a* in the HbChP from E12.5 to E17.5 (Fig. 3a), with its expression being restricted to HbChP epithelium at E13.5 (Fig. 3b), while in TelChP *Wls* transcripts can be detected only in the adjacent CH region (Fig. 3b, asterisk). At a protein level, WLS was readily detected in the HbChP, but not in the TelChP, as assessed by western blot at E14.5 and E17.5 using a validated WLS antibody (Fig. 3c and Supplementary Fig. 4a). Immunofluorescence analysis showed that while WLS was absent from the TelChP at E14.5 (Fig. 3d), high WLS levels were detected in the HbChP epithelium. WLS was also detected in the CH adjacent to the TelChP (Fig. 3d, asterisk). Notably, both *Wls* expression and its protein abundancy were highest in HbChP epithelial cells that were most positive for *Wnt5a* (Fig. 3e, f, arrowheads). Levels of WLS and WNT5A showed similar dynamics with high levels at E12.5 and E14.5, by E17.5 signal intensity decreased and became undetectable by postnatal day 23 (Fig. 3d and Supplementary Fig. 4b). Thus, our results are compatible with a possible role of the HbChP in secreting WNT5A into the CSF during embryonic development.

**Active WNT5A is secreted by HbChP cells.** To address this possibility, we first examined whether WNT5A is present in the embryonic CSF at E11.5, before the formation of the HbChP[23], and after that, at E14.5. Western blot analysis revealed the presence of WNT5A in the CSF at E14.5, but not at E11.5 (Fig. 4a). We next established a primary culture of E14.5 ChP epithelial cells and examined their capacity to secrete WNT5A to the media. Both TelChP and HbChP cells were positive for the ChP epithelial markers, ZO1[24] and AQP1 (Fig. 4b). The analyses of media conditioned by these cells showed the presence of WNT5A in media from the HbChP, but not by the TelChP primary cells (Fig. 4c). These results indicate that WNT5A is indeed produced and secreted from primary HbChP epithelial cells. Moreover, we found that the conditioned medium (CM) obtained from the HbChP, but not from the TelChP, was able to activate Wnt signaling, as shown by the capacity of HbChP-derived CM to induce phosphorylation of Dishevelled-3 (DVL3) at a level comparable to that of recombinant WNT5A (Fig. 4d). Thus combined, our data indicate that the WNT5A secreted by epithelial cells from the HbChP is biologically active. To determine whether the

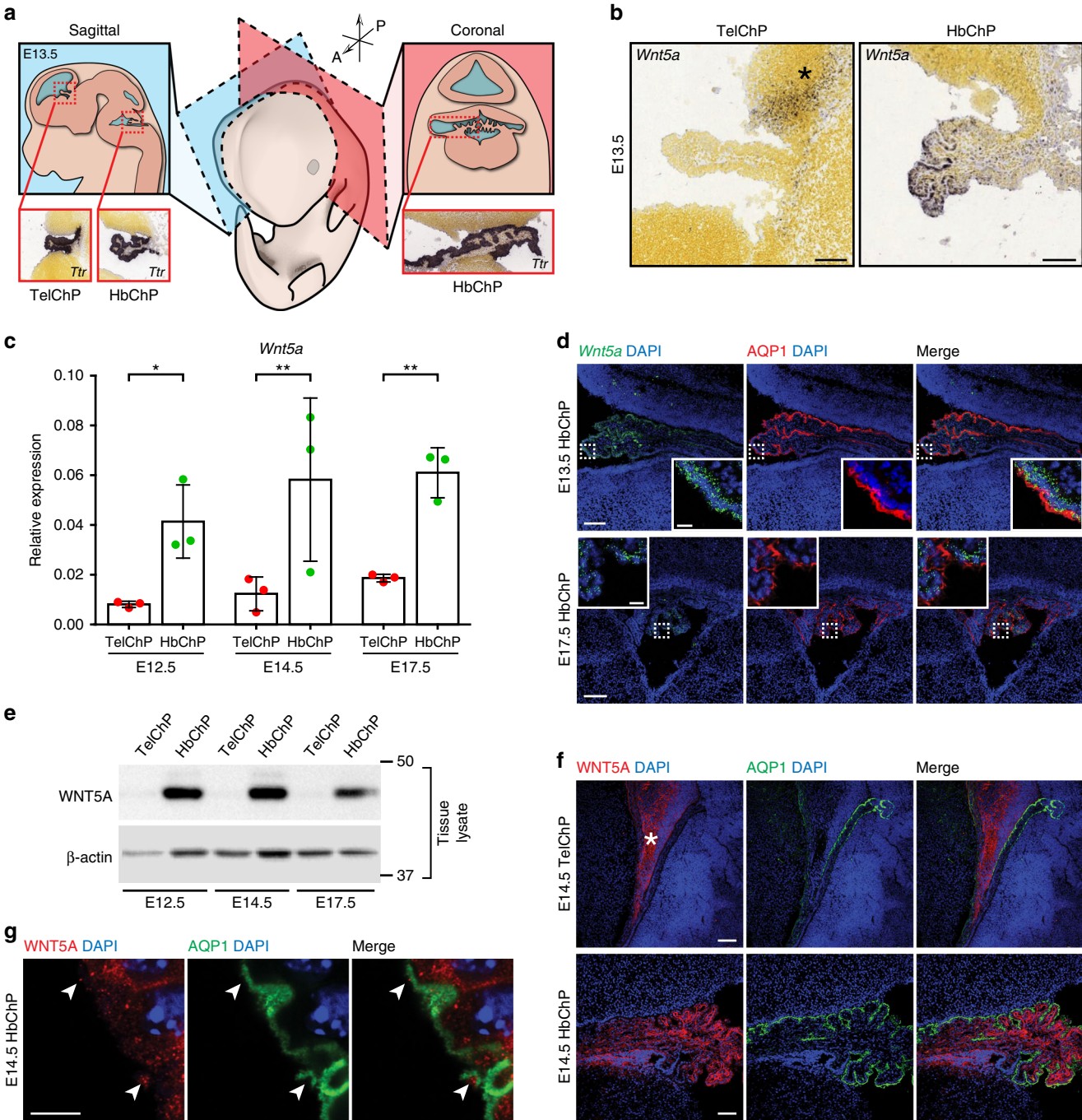

**Fig. 1** Wnt5a expression is restricted to HbChP. **a** Localization of TelChP and HbChP within the embryonic brain and illustrative pictures of in situ hybridization for ChP marker *Transthyretin* (*Ttr*) in sagittal sections of TelChP and HbChP and coronal section of HbChP. Image credit: Allen Institute. *A* anterior, *P* posterior. **b** Sagittal sections showing *Wnt5a* expression being restricted to HbChP epithelium and absent from the stromal cells at E13.5. *Wnt5a* expression is absent from the TelChP, but present in CH (asterisk). Image credit: Allen Institute. Scale bar: 100 μm. **c** Real-time qPCR of *Wnt5a* expression in TelChP and HbChP at E12.5, E14.5 and E17.5. The expression was normalized against expression level of *β-actin* in each condition. Graph shows $n = 3$ biologically independent samples; error bars represent mean ± s.d.; *P*-values (two-tailed Student's *t*-test with unequal variance): * $P < 0.05$, ** $P < 0.01$. TelChP vs HbChP: E12.5 $P = 0.012$; E14.5 $P = 0.0011$; E17.5 $P = 0.0023$. Biological replicates are indicated in the graph. **d** In situ hybridization and immunostaining analysis of HbChP coronal sections shows *Wnt5a* expression level differences between E13.5 and E17.5, $n = 3$. Immunostaining of Aquaporin-1 (AQP1) was used as a marker of HbChP epithelial monolayer highlighting the apical membrane of the tissue. Inset: Magnified view of fluorescent signal of *Wnt5a* transcripts detected in HbChP epithelium. Scale bar: 50 μm, inset scale bar: 10 μm. **e** Western blot analysis of WNT5A protein in lysates of TelChP and HbChP at E12.5, E14.5 and E17.5, $n = 3$. β-actin serves as a loading control. **f** Immunofluorescence analyses of WNT5A in TelChP and HbChP at E14.5, $n = 4$. Comparison with the AQP1 shows absence of WNT5A in the TelChP. In contrast, WNT5A can be detected in the HbChP epithelial layer. Scale bar: 100 μm. **g** High magnification view of HbChP epithelium apical membrane highlighted by AQP1 showing WNT5A in punctate structures in the extracellular domain of the plasmatic membrane (arrowheads); $n = 4$. Scale bar: 5 μm. Source data are provided as a Source Data file

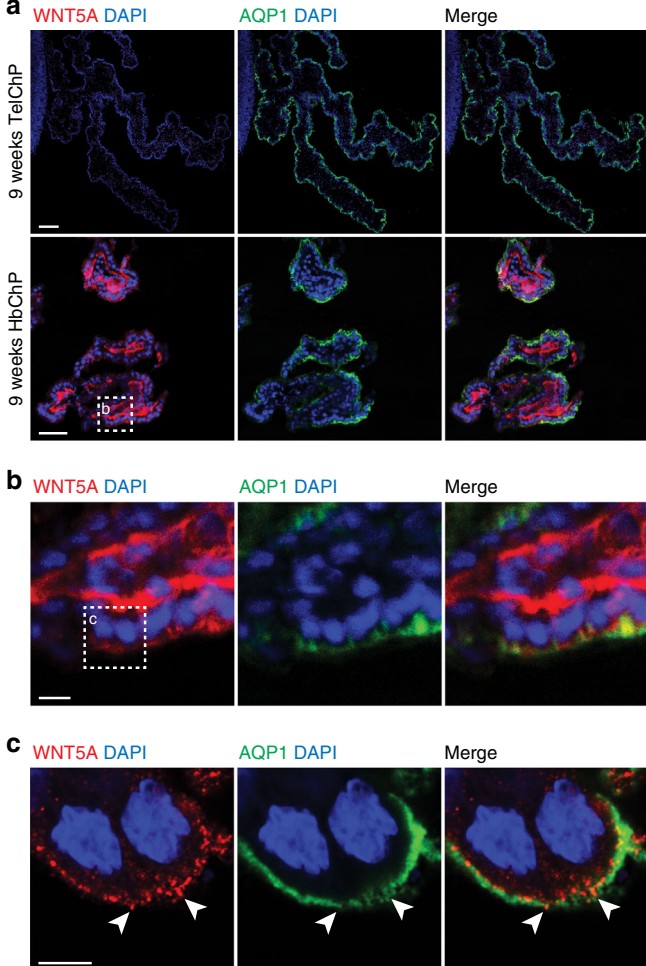

**Fig. 2** WNT5A is restricted to HbChP in human embryos.
**a** Immunofluorescence analysis of WNT5A in 9 weeks old human fetal samples shows the absence of signal in TelChP, while WNT5A can be readily detected in HbChP, n = 3. AQP1 is a marker of apical part of ChP epithelial cells. Scale bar: 200 μm. **b** Magnified portion of HbChP epithelial layer. Scale bar: 50 μm. **c** Magnified view of picture in 2b, showing the apical membrane of HbChP epithelium highlighted by AQP1, demonstrates presence of WNT5A in the punctate structures located at the cell surface (arrowheads). Scale bar: 2 μm

activation of Wnt signaling is mediated by WNT5A and not by any other Wnt or factors secreted by the HbChP primary culture, we generated a *Wnt5a* conditional knock-out mice (*Wnt5a^{cKO}*) by crossing *FoxJ1-CreERT2* and *Wnt5a^{fl/fl}* mice (Fig. 4e, f). We confirmed that WNT5A was completely absent in the HbChP (Fig. 4g, h), while it was still present in other embryonal regions (Fig. 4i), validating efficacy of our strategy. We also demonstrated a complete absence of WNT5A in the media from embryonic HbChP primary cultures obtained from *Wnt5a^{cKO}* embryos at E14.5 (Fig. 4j). Moreover, media derived from these cultures failed to elicit any activation of Wnt signaling, compared to medium obtained from wild-type (*WT*) HbChP primary cultures (Fig. 4k). We also found that the HbChP media activated only readouts for Wnt/β-catenin independent signaling such as phosphorylation of the receptor tyrosine kinase-like orphan receptor 1 (ROR1) or DVL2/DVL3 (similarly to recombinant WNT5A), but not unique readouts for Wnt/β-catenin signaling such as downregulation of Axin1 protein level, accumulation in the active β-catenin or induction of β-catenin target genes *Tcf1* and *Axin2*. All these readouts could be, however, efficiently

induced by recombinant WNT3A or Wnt3a CM in the same cells (Fig. 4k and Supplementary Fig. 5a, b). Altogether these results indicate that WNT5A and no other WNTs are the source of the WNT signaling activity in the media produced by HbChP primary cells and that such HbChP-produced WNT5A activates Wnt/β-catenin independent signaling.

**WNT5A associates with lipoproteins in the HbChP epithelium.** Previous studies have described the association of WNTs to either exosomes or lipoproteins in different cellular models[25,26]. Extracellular vesicles, such as exosomes, and lipoprotein particles can both be separated from cell supernatants by differential centrifugation. We thus fractionated mouse HbChP supernatants by ultracentrifugation and characterized the different fractions (Fig. 5a and Supplementary Fig. 6a). Immunoblot analysis showed that WNT5A is not present in the fraction containing exosomes and positive for various exosomal markers, such as CD63, HSP70, tumor susceptibility gene 101 (TSG101), and Flotillin-2 (FLOT2). Instead, WNT5A was enriched in the supernatant, a fraction positive for Apolipoprotein E (APOE) and Apolipoprotein J (APOJ, also known as Clusterin), which are integral structural components of lipoprotein complexes (Fig. 5b). In support of this finding, immunofluorescence analysis of E14.5 mouse embryos revealed that WNT5A does not colocalize with any exosomal marker (Fig. 5c, arrowheads).

We next decided to investigate the possible association of WNT5A to lipoprotein particles. Lipoproteins represent a heterogeneous family of macromolecule complexes classified by their size, lipid-to-protein ratio and presence of specific apolipoproteins[27]. Immunofluorescence showed a high degree of colocalization between extracellular WNT5A puncta and different species of apolipoproteins including Apolipoprotein A-I (APOA1), Apolipoprotein B-100 (APOB) and APOE in HbChP epithelium (Fig. 5d). Quantitative analysis of the colocalization of WNT5A and apolipoproteins or exosomal markers in the apical portion of HbChP epithelium (see methods and Supplementary Fig. 7a) revealed that 20–30% of WNT5A colocalized with apolipoproteins: 33.25% with APOJ, 20.77% with APOA1, 31.28% with APOB, 27.19% with APOE, while only 3-7% co-localized with exosomal markers—CD63 and TSG101 (Fig. 5e). The colocalization of WNT5A and apolipoproteins was specific as shown by the near complete absence of colocalization of WNT5A+ and APOA1+ puncta in the *Wnt5a^{KO}* compared to *Wnt5a^{WT}* HbChP epithelium (Supplementary Fig. 7b) as well as by additional controls of antibody specificity (Supplementary Fig. 7c and 8a–i). Furthermore, quantitative analysis of the colocalization of WNT5A with APOJ, an apolipoprotein highly enriched in the central nervous system (CNS)[28], and other apolipoproteins revealed extensive triple co-localization with APOA1 (71.93%), APOB (48.72%), or APOE (49%) (Fig. 5f). In contrast, there was almost no triple colocalization of APOJ, WNT5A, and the exosome marker CD63 (Fig. 5g). From these data we conclude that WNT5A is mainly found in association with apolipoproteins in epithelial cells of the HbChP.

**Extracellular WNT5A associates with lipoproteins.** To study the association of WNT5A and different classes of lipoprotein particles outside of the ChP cells, we performed discontinuous gradient ultracentrifugation of HbChP-derived primary culture supernatants. WNT5A was found to segregate both with the low-density lipoprotein (LDL) fraction and the high-density lipoprotein (HDL) fraction, (Fig. 6a and Supplementary Fig. 9a). In addition, mouse V5-tagged WNT5A (WNT5A-V5) co-immunoprecipitated HA-tagged APOJ or APOE (Fig. 6b, c, asterisk) in HEK293T cells. Likewise, APOE or APOJ were able to

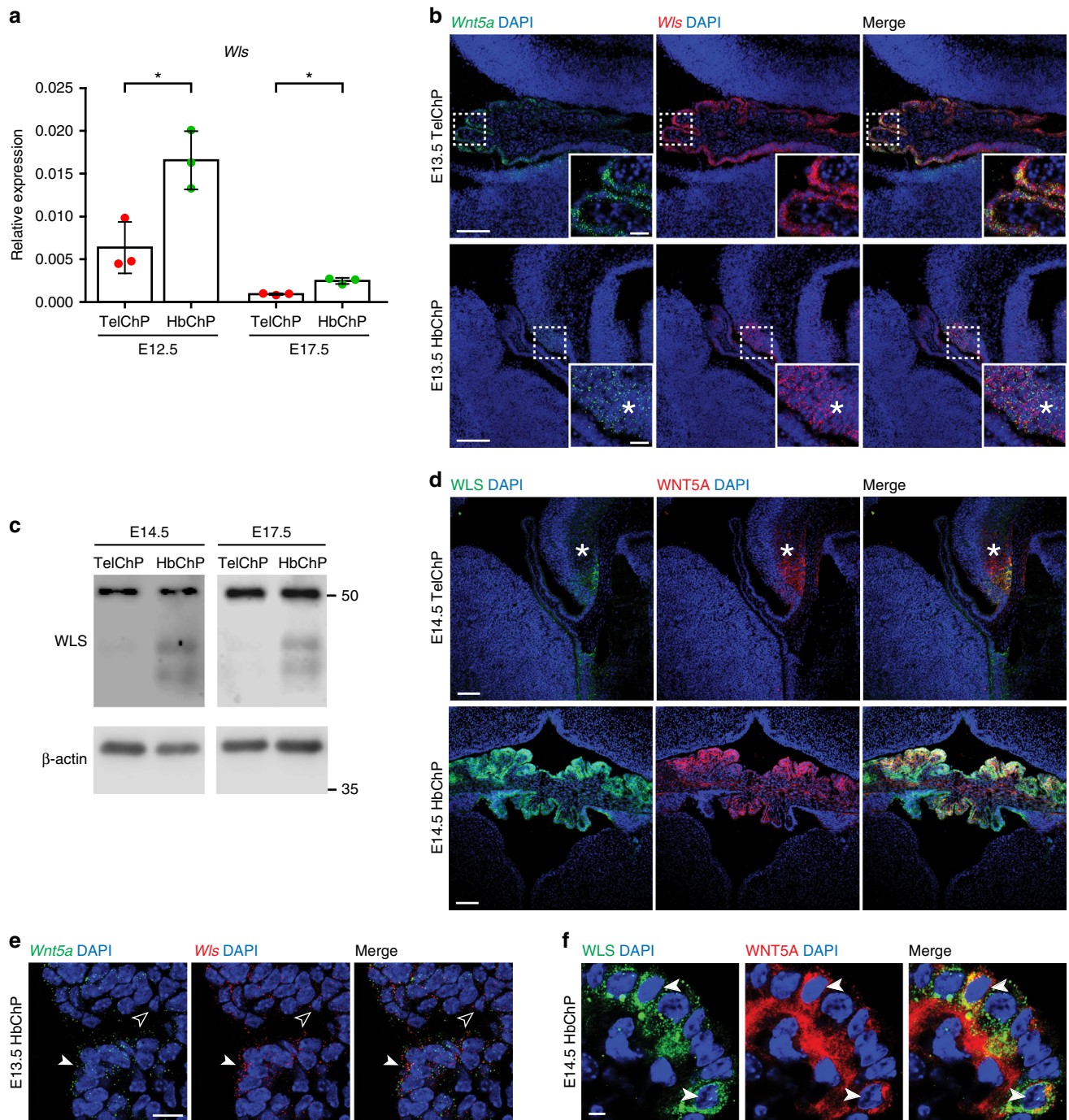

**Fig. 3** Wls is specifically expressed and produced in HbChP. **a** Gene expression analysis by real-time qPCR of Wntless (Wls) expression in TelChP and HbChP at E12.5 and E17.5. Wls gene expression was normalized against expression level of β-actin in each condition. Graph shows n = 3 biologically independent samples; error bars represent mean ± s.d.; P-values (two-tailed Student's t-test with unequal variance) for differences between analysed ChPs: * $P < 0.05$. TelChP vs HbChP: E12.5 $P = 0.0261$; E17.5 $P = 0.0237$. Biological replicates are indicated in the graph. **b** In situ hybridization analysis of HbChP coronal sections confirms Wls expression levels difference between E13.5 TelChP and HbChP, n = 6. Wls and Wnt5a expression in TelChP region is restricted mostly to CH (asterisk) and not to TelChP epithelium. Scale bar: 50 μm. Inset images show higher magnification view of fluorescent signal corresponding to Wls and Wnt5a transcripts in corresponding TelChP and HbChP epithelium. Scale bar: 10 μm. **c** Western blot analysis of WLS levels in tissue lysates of TelChP and HbChP isolated at E14.5 and E17.5, n = 3. β-actin serves as a loading control. **d** Representative images from immunofluorescent analysis of TelChP and HbChP at E14.5 displaying presence of WLS only in the HbChP, n = 4. WLS is absent from the TelChP, but present in CH (asterisk). Scale bar: 100 μm. **e** Detailed view of Wnt5a and Wls transcripts distribution highlights correlation between levels of Wls and Wnt5a in individual HbChP epithelial cells at E13.5 (arrowhead - $Wnt5a^{high}/Wls^{high}$; empty arrowhead - $Wnt5a^{low}/Wls^{low}$), n = 6. Scale bar: 5 μm. **f** Magnified image of immunofluorescent analysis of HbChP epithelium at E14.5 shows high degree of correlation between WNT5A and WLS levels observed within HbChP epithelial cells (arrowheads), n = 4. Scale bar: 5 μm. Source data are provided as a Source Data file

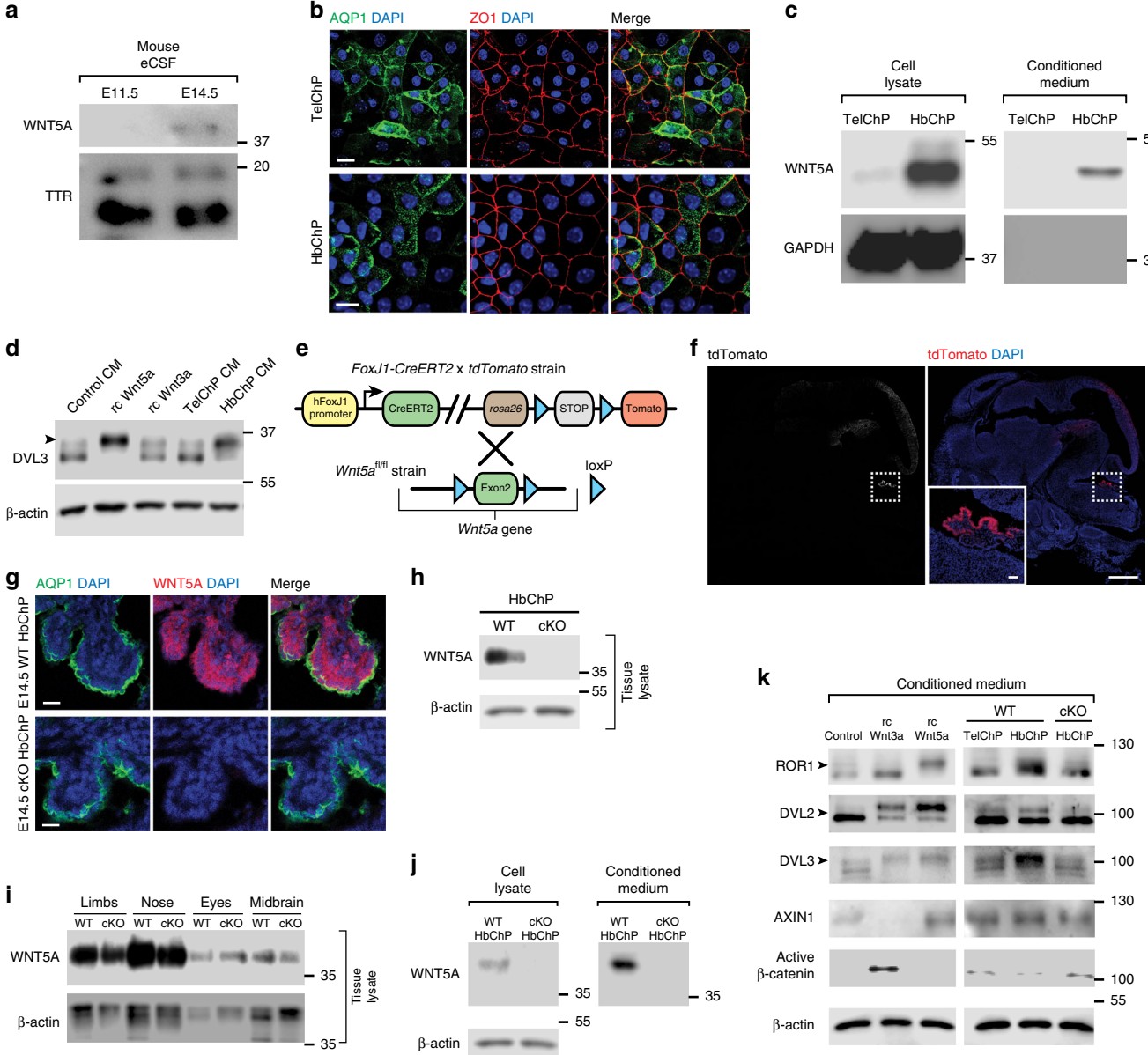

**Fig. 4** WNT5A is secreted by the HbChP epithelial cells. **a** Western blot of WNT5A in mouse embryonic CSF isolated at E11.5 and E14.5. CSF marker TTR used as a loading control, n = 3. **b** Representative image of immunofluorescent analysis of primary cultures of E14.5 TelChP and HbChP epithelial cells, n = 4. AQP1 and ZO1 - marker of tight junctions. Scale bar: 10 μm. **c** Western blot detection of WNT5A in cell lysate and supernatants from TelChP and HbChP primary cultures characterized in **b**, n = 4. GAPDH serves as a loading control (cell lysate) and as a control for cell contamination (supernatant). **d** Supernatants from HbChP epithelium primary culture (E14.5) can activate Wnt pathway as demonstrated by western blot analysis of DVL3 shift (phosphorylation, see arrowheads) in MEF cells treated with the supernatant, n = 3. Loading control: β-actin. Negative control: 10% FBS DMEM medium; positive control: recombinant WNT5A (rcWnt5a, 200 ng/ml), recombinant WNT3A (rcWnt3a, 200 ng/ml). **e** Schematic representation of mouse strains used to generate conditional *Wnt5a*cKO in the ChP. **f** Immunostaining images of sagittal section from E14.5 *FoxJ1-CreERT2-tdTomato* embryos with tamoxifen injection at E10.5. Inset: recombination in HbChP epithelium tracked by tdTomato, n = 3. Scale bar: 500 μm, inset scale bar: 50 μm. **g** Immunostaining of coronal section of HbChP epithelium in *Wnt5a*WT and *Wnt5a*cKO mouse embryo at E14.5, n = 4. Signal for WNT5A is completely absent from *Wnt5a*cKO HbChP epithelium as compared to *Wnt5a*WT littermate. Scale bar: 20 μm. For g-k: Tamoxifen injection at E12.5. (**h, i**) Western blot for WNT5A in tissue lysates at E14.5. WNT5A is missing in *Wnt5a*cKO HbChP **h** but not in the other analysed tissues **i**, n = 3. Loading control: β-actin. **j** Primary cultures of *Wnt5a*cKO-HbChP do not produce and secrete WNT5A, n = 3. Loading control: β-actin. **k** *Wnt5a*cKO-HbChP primary culture CM fails activate Wnt pathway in MEF cells. MEF cells were treated for 4 h as indicated and Wnt pathway signalling components activation was assessed by western blotting, n = 3. Phosphorylation-dependent shift in the readouts of WNT5A-triggered signalling (ROR1, DVL2, and DVL3) is indicated by the arrowheads. Positive control: recombinant WNT5A (rcWnt5a, 200 ng/ml), recombinant WNT3A (rcWnt3a, 200 ng/ml). Loading control: β-actin

pull-down WNT5A-V5 indicating that WNT5A and APOE or APOJ can form a complex (Fig. 6d, e). In addition, WNT5A-V5 was able to pull-down endogenous APOJ from the lipoprotein fraction isolated from the media of TR-CSFB cells, a ChP-derived cell line[29], (Supplementary Fig. 9b). Moreover, mass spectrometry

analysis of a WNT5A-V5 pull-down from TR-CSFB cells CM identified enrichment in additional proteins commonly associated with the HDL-specific proteome, such as Apolipoprotein A-I, Apolipoprotein A-II or Vitamin D-binding protein (Supplementary Fig. 9c). Strengthening these observations, we also

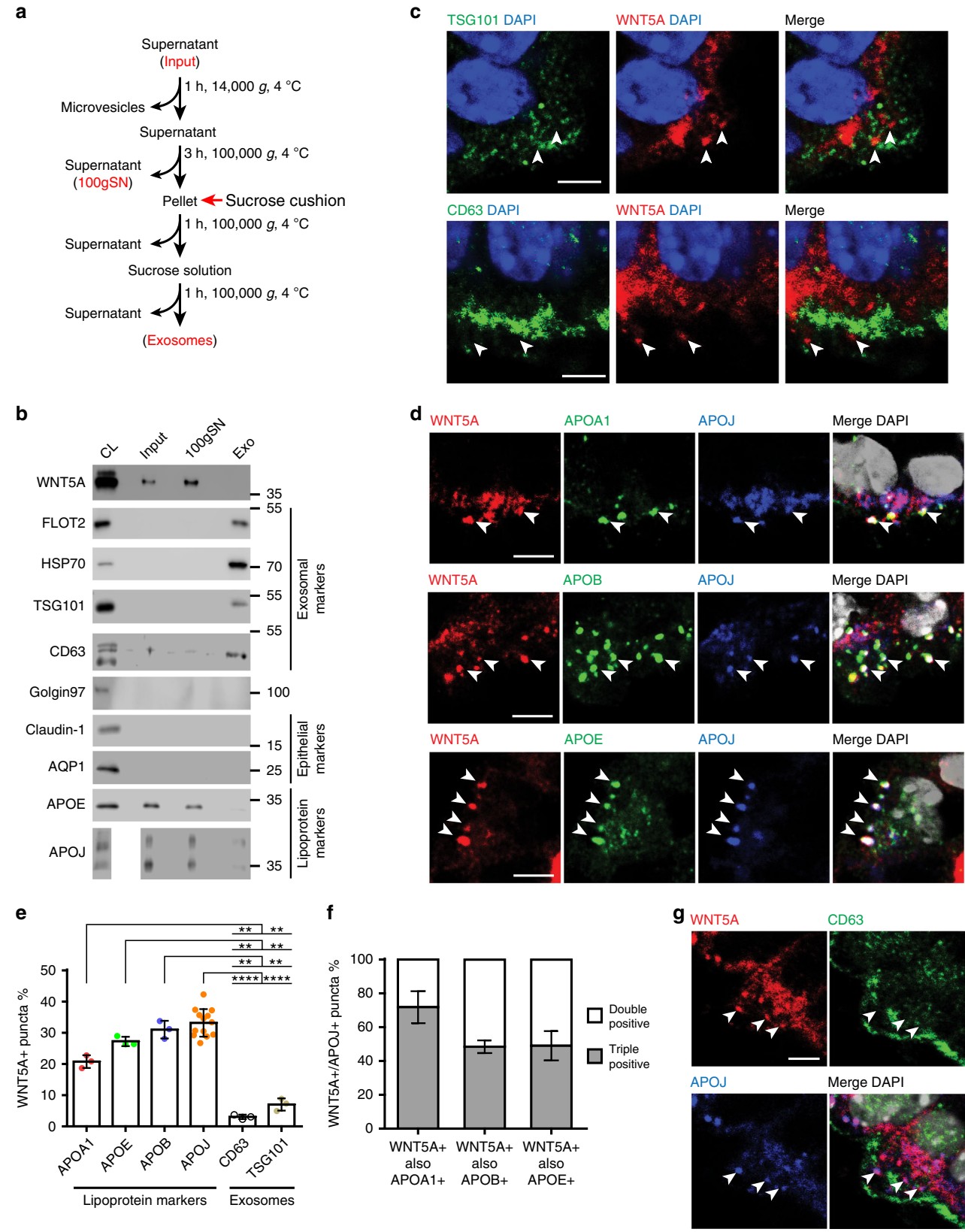

demonstrate direct in vitro interaction of WNT5A with human APOJ as shown via co-immunoprecipitation analysis (Fig. 6f).

To further investigate the necessity of lipoprotein particles for WNT5A transport, we used a lipid removal agent (LRA)[30] to delipidate the serum used in primary HbChP epithelial cell cultures (Fig. 6g). Upon lipid removal, WNT5A was not detected in the supernatants of primary HbChP epithelial cultures, while it slightly increased in cell lysates, suggesting that the ligand was

**Fig. 5** WNT5A is secreted and associated with lipoproteins in HbChP epithelium. **a** Schematic representation of the differential centrifugation protocol used for the isolation of exosomes and enrichment of lipoproteins from primary culture supernatants. **b** Western blot analysis of HbChP epithelial primary culture CM (Input), lipoprotein (100gSN) and exosomal (Exo) fractions, n = 3. Exosomal markers: FLOT2, HSP70, TSG101, CD63; lipoproteins: APOE and APOJ; negative control: Golgin97 (marker of Golgi) and epithelial markers: Claudin-1 and AQP1. **c, d** Immunofluorescent analysis of co-localization between markers of exosomes - CD63 and TSG101 **c** or various apolipoproteins **d** with WNT5A in E14.5 in HbChP epithelium, n = 3. WNT5A puncta only poorly overlap with exosomal markers CD63 and TSG101 (arrowheads) but show significant degree of overlap (arrowheads) with APOA1, APOB, APOE and APOJ. Scale bar: 5 μm. **e** Quantification of co-localization between WNT5A puncta and apolipoproteins (APOA1, APOB, APOE, APOJ) or exosomal markers (TSG101 and CD63). Graph shows n = 3 biologically independent samples. 3 representative images from 3 consecutive sections – 1 image per section - have been analyzed for each combination of markers and graphs show mean ± s.d. Quantification has been performed by IMARIS software and differences analysed by two-tailed Student's t-test with unequal variance (*P < 0.05; **P < 0.01; ****P < 0.0001). WNT5A+/APOs+ puncta vs WNT5A+/CD63+ and WNT5A+/TSG101+, respectively: APOA1+ $P = 0.0076$ and $P = 0.009$; APOE+ $P = 0.0014$ and $P = 0.009$; APOB+ $P = 0.0048$ and $P = 0.0094$, APOJ+ $P = <0.0001$ and $P = <0.0001$. **f** Quantification of co-localization between puncta double-positive for WNT5A and APOJ which are also positive for APOA1, APOB and APOE, respectively. **g** Representative image directly illustrating the extent of immunofluorescent signal overlap between WNT5A and either APOJ as a marker of lipoproteins and CD63 as a marker of exosomes, n = 3. Scale bar: 5 μm. Source data are provided as a Source Data file

retained inside the cells (Fig. 6h). On the other hand, when cells grown in delipidated serum for 3 days, were treated with serum (which does not contain WNT5A), WNT5A was again found in the media (Fig. 6h). Moreover, treatment of delipidated cultures with HDL or LDL fractions isolated from mouse serum rescued the presence of WNT5A in the media (Fig. 6i), indicating that lipoproteins are sufficient to restore the presence of soluble WNT5A in the media of HbChP epithelium primary cultures. In addition, in vitro experiments showed that recombinant WNT5A physically binds to LDL and forms a high-molecular weight complex (Fig. 6j, k) that retains capacity to activate Wnt signal transduction (Fig. 6l, m). Thus, our results indicate that apolipoproteins are required for the secretion of WNT5A by HbChP cells and that the WNT5A-lipoprotein complexes are functional.

**WNT5A derived from HbChP regulates hindbrain morphogenesis.** After demonstrating that active WNT5A is secreted into the CSF by epithelial cells of the HbChP, we focused our attention on the possible effect of HbChP-derived WNT5A on target cells in vivo. For our analysis we selected the ventricular zone of the developing cerebellum since it is in the proximity of the HbChP, and it is exposed to high local concentrations of HbChP-derived factors, as previously described for Shh[31]. Dorsal hindbrain progenitor cells lining the ventricle, anterior to the HbChP, devoid of *Wnt5a* expression at E13.5 (empty arrowheads in Fig. 7a) were found to express WNT receptors, such as the *Fzd3* and *Fzd10*, and WNT/PCP signaling components, such as *Vangl2* or *Celsr2* (Supplementary Fig. 10a, arrowheads)[32]. In addition, lipoprotein receptors such as *Scarb1*[33], *Lrp2*[34], and *Lrp4*[35], receptors previously shown to be involved in Wnt signaling[36], are also expressed in this region (Supplementary Fig. 10b). Moreover, while WNT5A protein was found in the apical surface of *WT* hindbrain progenitors, colocalizing with APOE and APOJ, it was not found in the *Wnt5a*KO embryos (Fig. 7b, c). These findings support the hypothesis that WNT5A/apolipoprotein complexes target hindbrain progenitor cells lining the ventricular cavity. We thus decided to investigate the developing cerebellum and examine whether WNT5A produced by HbChP cells can control developmental processes typically regulated by WNT5A, such as morphogenesis or the balance between cell proliferation and differentiation[37,38]. Analysis of either complete or conditional HbChP *Wnt5a* knock-out (*Wnt5a*KO or *Wnt5a*cKO) embryos revealed a robust reduction of the size (area) of the developing cerebellum (Fig. 7d–h). While the dorso-ventral axis in the midline decreased (Fig. 7i), no major difference was found in the latero-medial axis (Fig. 7j). Diminished tissue size can be thus described as a disproportionate shortening of the medial dorso-ventral axis and an increase in the width-to-length ratio (Fig. 7k).

Analysis of the proliferative markers in this region showed a significant increase in the number of KI67+ but not EdU+ cells in *Wnt5a*KO embryos at E16.5 (Supplementary Fig. 11a–d). However, no change in KI67+ cells was observed in *Wnt5a*cKO embryos at E16.5 (Supplementary Fig. 11e, f). Additionally, we did not observe any change in the number of activated caspase 3-positive cells in either *Wnt5a*KO or *Wnt5a*cKO embryos (Supplementary Fig. 11g, h). Altogether this indicates that changes in cerebellum morphology in the *Wnt5a*KO and *Wnt5a*cKO are not due to an alteration of proliferation or survival, but rather can be attributed to an aberrant morphogenesis.

## Discussion

In the present study, we demonstrate that WNT5A is produced in the epithelial cells of the HbChP in a very specific temporal developmental window, from E12.5 to early postnatal age in mouse. Likewise, analysis of human fetal brains at week 9 confirmed the presence of WNT5A in the HbChP. Our results show that WNT5A colocalizes with lipoproteins in the apical part of both the HbChP epithelium, a cell type that expresses *Wnt5a*, and in target hindbrain progenitors lining the ventricle, that express Wnt receptors, signaling components as well as receptors for lipoprotein particles, but do not express *Wnt5a*. Moreover, we found that the HbChP secretes WNT5A, which associates with lipoprotein particles in the CSF. We thus suggest that lipoprotein particles provide a vehicle for the transport of WNT5A through the CSF, from the HbChP to target cells lining the ventricle. In agreement with this possibility, we found that *Wnt5a* is required for morphogenesis of dorsal hindbrain. This observation is in line with previous reports highlighting the pivotal role of Wnt5a in morphogenesis of various regions of the developing CNS, including the ventral midbrain[39] and the cerebellum[40]. However, unlike the developing ventral midbrain[41], we did not observe an effect on proliferation in the *Wnt5a*cKO, or on cell survival, suggesting a selective alteration of morphogenesis.

With regards to the mechanism of the transport of WNT5A in the CSF, several lines of evidence support WNT5A being transported via lipoproteins rather than exosomes. Firstly, immunofluorescence analysis indicates that WNT5A mainly colocalizes with apolipoproteins rather than with exosomal markers in the apical part of the HbChP epithelium. Second, biochemical fractionation of media from HbChP cultures showed that WNT5A segregated with apolipoproteins, but not with exosomal markers. Third, WNT5A was also found to co-localize with apolipoproteins in target cells lining the ventricle, highlighting the role of lipoproteins in the transport of WNT5A from its source to target cells. Notably, we found that WNT5A interacts with several apolipoproteins, such as APOA1, APOB, APOE, and APOJ, a result in agreement with the previous identification of all major

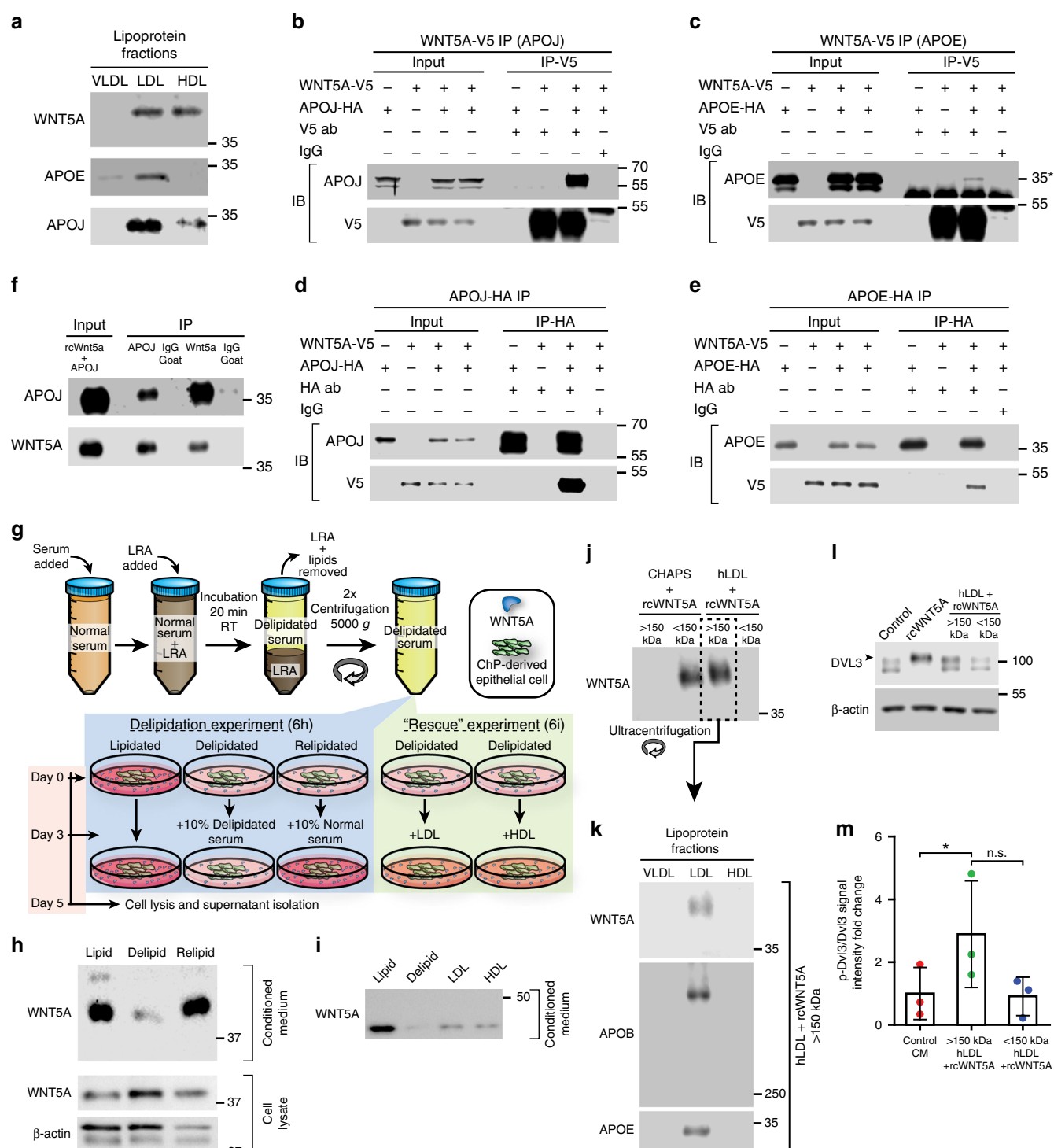

classes of lipoprotein particles in CSF by proteomic analyses[42–45]. This is also in agreement with the putative role of the ChP as the main entry point for lipoprotein particles into the CSF for their distribution into the CNS[46].

Our study also identifies WNT5A as an additional member of a growing family of biologically active signaling molecules produced in the ChP that are present in the CSF[47], and are transported in lipoprotein particles[48]. Moreover, WNT5A shares with another lipid-modified morphogen, SHH, the fact that it is produced in the HbChP, distributed in a transventricular fashion, transported via lipoproteins and capable of regulating neural development in progenitor cells lining the ventricular

cavity[31,49,50]. On the other hand, lipoproteins such as LDL, have been found to contribute to the capacity of the CSF to control neural development and promote neuronal differentiation[51]. Interestingly, we found expression of various receptors for lipoprotein particles within the ventricular zone of developing brain, which combined with the previously reported accumulation of lipoprotein particles in the ventricular zone of the brain[46], suggests a possible role for lipoproteins complexes to act as a universal vehicle for distribution of lipids and lipid-modified signaling molecules within the CNS. Our data suggest that one of the mechanisms by which lipoproteins may control neural development is by associating to and transporting signaling

**Fig. 6** WNT5A is present in lipoprotein complexes. **a** CM from HbChP primary culture was subjected to KBr gradient-based ultracentrifugation and different isolated fractions (VLDL, LDL, and HDL) were analysed for the presence of WNT5A, APOE and APOJ by western blot, $n = 3$. (**b–e**) WNT5A-V5 and HA-tagged APOJ (**b, d**) or APOE (**c, e**) were overexpressed in HEK293T cells. WNT5A-V5 (**b, d**) and APOJ/E (**c, e**) were immunoprecipitated using anti-V5, anti-HA antibody and control IgG, respectively, and detected by western blotting, $n = 3$. Input is the loading control. Asterisk indicates non-specific immunoglobulin light chain. **f** Recombinant WNT5A (rcWNT5A) interacts with isolated APOJ protein. WNT5A and APOJ co-immunoprecipitated together. IgG served as a control, $n = 3$. Input represents the initial mixture. **g** Schematic depiction of the experimental design for delipidation and rescue experiments. **h** Delipidated FBS (Delipid) prevents production of WNT5A that is restored by relipidation (Relipid). Presence of WNT5A in cell lysates and CM from primary HbChP cultures has been analyzed using western blot, $n = 3$. Loading control: β-actin. **i** WNT5A secretion was restored after the addition of different mouse lipoprotein fractions to primary HbChP cultures cultivated in presence of delipidated conditions. **j** Recombinant rcWNT5A was incubated with either 0.6% CHAPS or human LDL (hLDL)- lipoprotein fraction for 4 h at 37 °C and filtered through 150 kDa cut-off protein concentrator. Western blot confirmed presence of WNT5A only in >150 kDa fraction but not in the <150 kDa fraction, $n = 3$. **k** Separation of >150 kDa fraction into VLDL, LDL, and HDL fractions using KBr gradient confirmed presence of rcWNT5A in LDL and its co-fractionation with APOB and APOE specific for LDL fraction, $n = 3$. **l** Only >150 kDa fraction of LDL/rcWNT5A mixture can trigger DVL3 phosphorylation (indicated by arrowhead) in MEF cells, $n = 3$. Negative control: 10% FBS DMEM; positive control: rcWNT5A; loading control: β-actin. **m** Statistical analysis of DVL3 shift assay. Graph shows $n = 3$ biologically independent samples; error bars represent mean ± s.d.; $P$-values (two-tailed Student's $t$-test with unequal variance): * $P < 0.05$. Control CM vs >150 kDa hLDL + rcWNT5A: $P = 0.0231$. Biological replicates are indicated in the graph. Source data are provided as a Source Data

molecules such as WNT5A. Indeed, deletion of *Wnt5a* eliminated the presence of WNT5A associated to lipoprotein particles in the apical part of hindbrain progenitor cells lining the ventricle, indicating the necessity of WNT5A for lipoprotein particles to gain such localization. Notably, our results also show overlapping expression domains of lipoprotein receptors and Wnt ligands receptors such as Frizzled receptors in ventricular zone of developing cerebellum. Moreover, Syndecans, a class of heparan sulfate proteoglycans that bind Wnts and are implicated in regulating the formation of Wnt ligand gradients and Wnt signaling[52], can interact with lipoprotein particles[53,54]. Combined, our results and data in the literature support the notion that lipoproteins, WNT5A and their receptors are required for proper neural development.

In sum, our study provides key evidence supporting the association of Wnts with lipoprotein particles, not only in WNT-producing cells of the ChP, but also after their secretion to the CSF and in target cerebellar progenitor cells lining the ventricular cavity. Our results thus support the hypothesis that the incorporation of Wnts, e.g., WNT5A, to lipoprotein particles, provides a mean for their secretion, extracellular dispersal and cell targeting. It remains to be determined whether the association of WNT5A to lipoprotein particles is a specific mechanism restricted to the HbChP or part of a more general strategy used by different cells to transport Wnts.

## Methods

**Mouse strains**. Embryos were obtained from female mice of CD1 IGS mouse strain (*Crl:CD1(ICR)* mice, Charles River Laboratories, Germany), *Wnt5a*[tm1Amc] (referred to as *Wnt5a*[KO] in the article)[55] or newly generated conditional knock-out mouse strain described below (referred to as *Wnt5a*[cKO] in the article).

Mouse strain *Wnt5a*[tm1.1Krvl/J] (referred to *Wnt5a*[flox/flox] in this article)[56] was purchased from Jackson laboratories; *Foxj1*[tm1.1(cre/ERT2/GFP)Htg] (referred to as *FoxJ1-creERT2* in this article)[57] and *Gt(ROSA)26Sor*[tm14(CAG-tdTomato)Hze] (referred to as tdTomato in this article)[58] were shared with Karolinska Institutet, Sweden on collaboration agreement. Induction of conditional knock-out or tdTomato reporter was induced by single dose tamoxifen (Sigma) intra-peritoneal injection of pregnant female mice in concentration of 4.5 mg of tamoxifen dissolved in sterile sunflower oil per 20 g weight of mouse. Embryos were harvested at E14.5 or E16.5. All mice strains were housed, bred, and treated in accordance with protocols approved by the local ethical committees (Stockholm's Norra Djurförsöksektiska Nämnd - N158/15, N326/12 or Czech Centre for Phenogenomics (Institute of Molecular Genetics, CAS) and Central Commission for Animal Welfare of Ministry of Agriculture Czech Republic - PP-90-2015).

**Embryonic CSF isolation**. Embryonic CSF was isolated from embryos of pregnant CD1 IGS mice. Embryonic CSF was obtained by microaspiration from E11.5 and E14.5 embryos, using pulled tip glass microcapillary pipette glass. Needle was inserted into the fourth brain ventricle with embryo being placed on its side. CSF collected for the analysis was obtained from 2 or 3 entire litters depending on the embryonic stage (20–25 embryos). Samples were microscopically controlled for

presence of blood contamination and samples showing signs of the contamination were discarded.

**Choroid plexus epithelial cells primary culture**. ChP tissue was collected from E14.5 embryos isolated from sacrificed pregnant CD1 mice and choroid plexus epithelial cells (CPEC) were isolated from TelChP and HbChP. During isolation, extracted tissue was kept at room temperature (RT) in HBSS solution (Gibco). After isolation, extracted tissue was briefly centrifuged (200 g, 10 s at RT). Following aspiration of supernatant, 500 μl of 2 mg/ml solution of Pronase (Sigma–Aldrich) was added to the extracted tissue and incubated for 5 min at 37 °C. Solution was then transferred to DMEM/F-12 medium containing 10% FBS (Gibco) and centrifuged (300 g, 3 min at RT). Tissue was transferred to complete culture medium consisting of DMEM/F-12 supplemented with 10% FBS, 10 ng/ml EGF (Invitrogen), 20 μM cytosine arabinoside (Sigma) 50 U/ml penicillin, and 50 U/ml streptomycin. Cells were mechanically dissociated by 6–8 times forced passage through a 21-gauge needle, followed by gentle repeated resuspension with 200-μl pipette. Finally, cells were seeded onto laminin (Invitrogen) coated 24-well plates ($2–3 \times 10^5$ cells per well). To achieve higher purity of epithelial cells, adhering-off method has been applied to reduce fibroblast contamination. After the initial seeding, supernatant containing unadhered cells was transferred to new laminin coated well thus removing from culture fibroblasts characterized by higher adherence affinity.

In order to produce CM, CPEC primary cultures were maintained in complete culture medium. CM was collected every 48 h up to 10 days after seeding. Supernatant was subjected to sequential centrifugation steps of 200 g for 5 min (to remove viable cells), 1500 g for 10 min (to remove dead cells), and 6000 g for 15 min (to remove cell debris). Used reagents are listed in the Supplementary Table 1.

**Fetal tissue section**. Ethical approval allowing human fetal tissue acquisition and analysis was provided by the National Research Ethics Service Committee East of England—Cambridge Central, UK (ethics number 96/085).

**Cell culture and transfection**. HEK293T cells were seeded in complete DMEM medium containing 10% FBS, 2 mM L-glutamine, 50 U/ml penicillin, and 50 U/ml streptomycin (Gibco) on 10 cm dishes 24 h prior transfection. The cells were transfected with total of 5 μg of DNA at ~40% confluency in DMEM medium only. The transfection reaction mixture was prepared using OptiMEM (Gibco) and Lipofectamine 2000 (Invitrogen), with ratio of 1 μg DNA: 2 μl Lipofectamine 2000, followed by incubation with cells for 4–6 h. Afterwards the transfection medium was exchanged for the complete medium. Used constructs are listed in the Supplementary Table 2.

TR-CSFB cells were seeded in DMEM containing 10% FBS, 2 mM L-glutamine, 50 U/ml penicillin, and 50 U/ml streptomycin (Gibco) on 10 cm dishes 24 h prior transfection. The cells were transfected with total of 8 μg of DNA in ~70% density in DMEM medium only. The transfection reaction mixture was prepared using PEI (Sigma) with ratio of 1 μg DNA: 2.5 μl PEI, followed by incubation of cells for 3 h. Afterwards the transfection medium was exchanged for the complete DMEM medium. CM was collected after 48 h and same pre-clearing protocol as mentioned for primary culture medium has been applied to remove any cell contamination. Used constructs are listed in the Supplementary Table 2.

Wnt3a and Wnt5a CM was isolated either from L-Wnt3a (ATCC CRL-2647) or L-Wnt5a cells (ATCC CRL-2814) according to ATCC instructions.

**Western blotting**. For western blot, samples were subjected to SDS-PAGE, electrotransferred onto Hybond-P membrane (GE Healthcare), immunodetected using appropriate primary and secondary antibodies and visualized by ECL

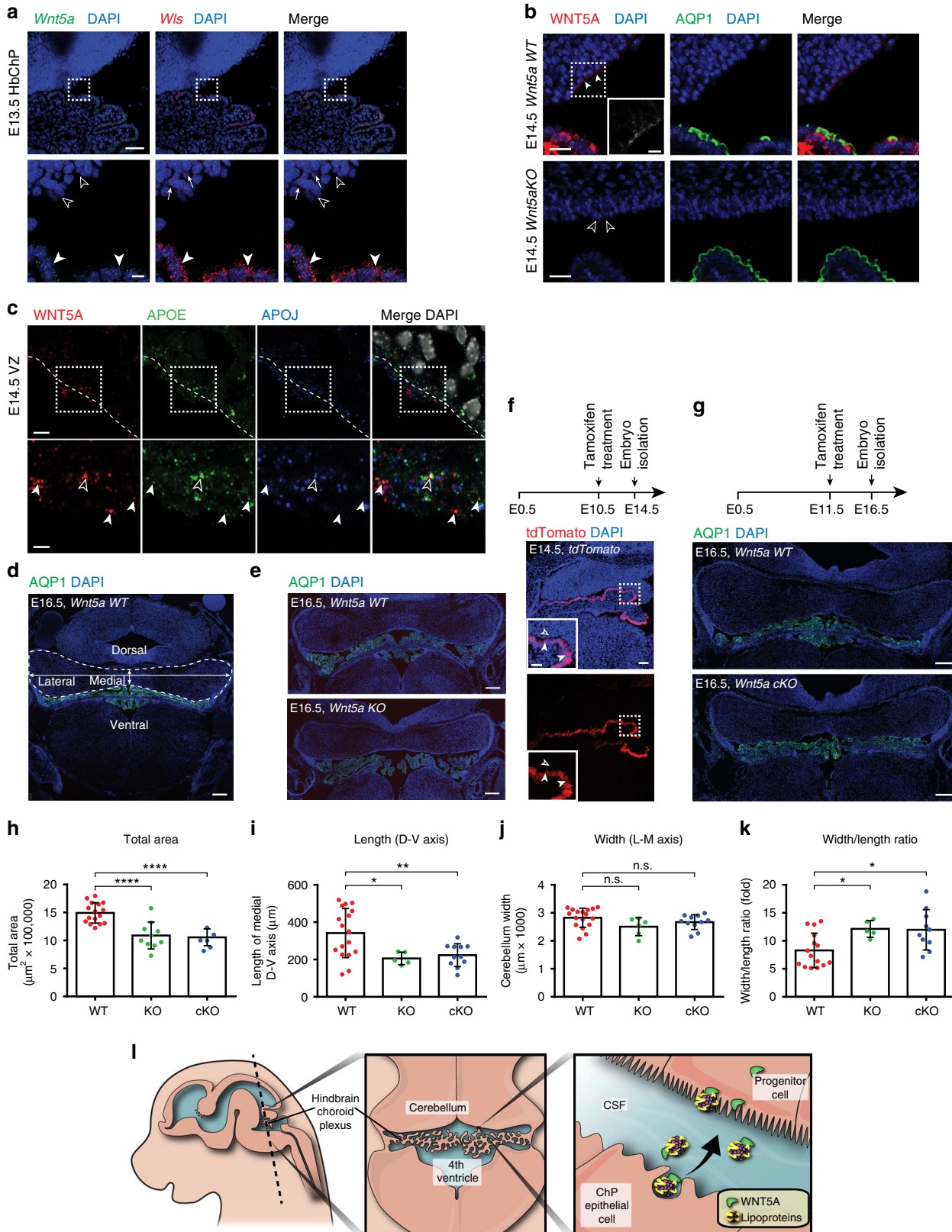

(GE Healthcare) or Supersignal Femto solution (Thermo Fisher). Western blot signal intensities were calculated using ImageJ. Briefly, area of the peak intensity for the protein of interest was divided by corresponding values of peaks intensity obtained for control protein. Uncropped images for all the western blots presented in the article can be found in Supplementary Fig. 12. Used reagents are listed in Supplementary Table 1. Used antibodies are listed in Supplementary Table 3.

**Co-Immunoprecipitation**. Dishes with transiently transfected HEK293T cells were placed on ice, washed twice with ice-cold PBS and lysed in 1 ml of lysis buffer for 20 min. Lysis buffer contained 50 mM Tris pH 7.6, 200 mM NaCl, 1 mM EDTA, 0.5% NP40, fresh 0.1 mM DTT (Sigma), and protease inhibitor cocktail (Roche). The lysis was centrifuged at 4 °C at 20,000 g for 20 min. After the centrifugation, an aliquot of the input sample representing 4% of the total cell lysate was taken from each condition. The lysates were then pre-cleared using 20 µl/sample DynaBeads

**Fig. 7** WNT5A secreted by HbChP controls morphogenesis of embryonic cerebellum. **a** In situ analysis of *Wnt5a* and *Wls* expression in the dorsal neuroepithelium adjacent to HbChP at E13.5 shows transcripts restriction to the HbChP epithelium, *n* = 4. Lower panel highlights absence of *Wnt5a* and low *Wls* expression in cerebellar ventricular zone (cVZ, empty arrowheads) compared to the HbChP (arrowheads). Scale bar: top 50 μm, bottom 10 μm. **b** WNT5A observed in the apical cVZ at E14.5 (arrowheads) is absent in *Wnt5a*^KO embryos (empty arrowheads), *n* = 3. Scale bar: top 5 μm, inset 2 μm. **c** WNT5A and APOE (arrowheads) or APOE with APOJ (empty arrowhead) immunostaining in the apical cVZ (dotted line) at E14.5, *n* = 4. Scale bar: top 5 μm, bottom 2 μm. (**d**) Scheme of analyzed cerebellar parameters: total area (dash line), width (latero-medial, L-M, horizontal arrows) and length (dorso-ventral, D-V, vertical arrows). Scale bar: 200 μm. **e** Coronal sections of cerebellum in *Wnt5a*^WT and *Wnt5a*^KO embryos. Scale bar: 200 μm. **f** tdTomato staining in E14.5 *FoxJ-CreERT2* embryo demonstrates recombination in HbChP epithelium, which is absent in the cVZ (inset, empty arrowhead) compared to HbChP (arrowheads), *n* = 3. Scale bar: top 50 μm, inset 20 μm. **g** Coronal sections of cerebellum in *Wnt5a*^WT vs *Wnt5a*^cKO embryos. Scale bar: 200 μm. **h**–**k** Analysis of the **h** total area, **i** length, **j** width **j**, and **k** width/length ratio of cerebellum in *Wnt5a*^WT, *Wnt5a*^KO and *Wnt5a*^cKO embryos. Graphs show individual data points (dots) from *n* = 3 biologically independent samples; error bars represent mean ± s.d.; *P*-values (two-tailed Student's t-test with unequal variance): * *P* < 0.05, ** *P* < 0.01, **** *P* < 0.0001. Corresponding *P*-values for differences between *Wnt5a*^WT (WT), *Wnt5a*^KO (KO) and *Wnt5a*^cKO (cKO)—**h**: WT vs. KO: 0.0004, WT vs. cKO: 0.0001; **i** WT vs KO: 0.0353, WT vs cKO: 0.0098; **j** WT vs. KO or cKO: – not significant; **k** WT vs KO: 0.016, WT vs cKO: 0.0115. **l** Schematic depiction of the model for WNT5A secretion by HbChP and its transventricular delivery to recipient regions. Source data are provided as a Source Data file

(Invitrogen) on a rotating shaker at 4 °C for 45 min. First, 400 μl of the pre-cleared samples was incubated with 1 μg of either anti-V5 (96025, Invitrogen), anti-HA (ab9110, Abcam) antibodies for 3 h and afterwards together with 30 μl DynaBeads for an additional 12–14 h. The samples were washed five times in 800 μl of lysis buffer (DTT- and protease inhibitors- free). Protein complexes were eluted with 50 μl of 1X Laemmli buffer in 95 °C for 5 min and were loaded directly onto a 10% acrylamide gel for western blot analysis.

In case of co-immunoprecipitation from supernatant, CM was collected after 48 h and same protocol as mentioned for primary culture medium was applied to remove any cell contamination. CM was next pre-cleared using 0.25 volume of G-protein Sepharose beads (GE Healthcare) and then incubated overnight with 0.5 μg/ml of anti-V5 antibody (cat.no. 96025, Invitrogen). Afterwards, the supernatant has been incubated with 50 μl of G-protein Sepharose beads. Resulting immunoprecipitates were washed several times with lysis buffer (DTT- and protease inhibitors-free) and twice in detergent-free lysis buffer. Protein complexes were eluted with 50 μl of 1X Laemmli buffer in 95 °C for 5 min and were loaded directly onto a 10% acrylamide gel for western blot analysis. Used reagents are listed in the Supplementary Table 1. Used constructs are listed in the Supplementary Table 2. Used antibodies are listed in the Supplementary Table 3.

**In vitro protein binding assay.** To assess physical interaction of WNT5A and apolipoproteins, 2 μg of human native APOJ (RD162034025, Biovendor) or human native APOE (cat.no. SRP6303, Sigma) were incubated with 150 ng of recombinant WNT5A (P22725, R&D Systems) in 200 μl of binding buffer (50 mM Tris pH 7.6, 100 mM NaCl, 1 mM EDTA) overnight at 4 °C. Next day, 50 μl of the mixture was incubated with 2 μg of corresponding antibody for several hours before the addition of 20 μl of DynaBeads followed by overnight incubation at 4 °C. Subsequently, solution containing the beads has been transferred to new tubes followed by quick wash three times with ice-cold binding buffer containing 0.5% NP40. After the last wash the beads were transferred again to a new tube. After aspiration of the remaining binding buffer, the beads were mixed with 1X Laemmli buffer and boiled for 2 min at 95 °C. Before loading onto gel, beads were removed from the 1X Laemmli buffer solution. Used reagents are listed in the Supplementary Table 1. Used antibodies are listed in the Supplementary Table 3.

**Mass spectrometry.** Beads with extracted proteins were washed three times with 50 mM ammonium bicarbonate buffer and subjected to digestion by trypsin (2 h, 37 °C; sequencing grade, Promega). Tryptic peptides were extracted into LC-MS vials by 2.5% formic acid (FA) in 50% acetonitrile with addition of polyethylene glycol (20,000; final concentration 0.001%) and concentrated in a SpeedVac concentrator (Thermo Fisher Scientific).

LC-MS/MS analyses of tryptic peptide mixture were done using RSLCnano system connected to Orbitrap Elite hybrid mass spectrometer (Thermo Fisher Scientific). Prior to LC separation, tryptic digests were desalted using trapping column (100 μm × 30 mm, 3.5 μm X-Bridge BEH 130 C18 sorbent, Waters) and separated on Acclaim Pepmap100 C18 column (2 μm particles, 75 μm × 500 mm; Thermo Fisher Scientific) by 1 or 2 h gradient program (mobile phase A: 0.1% FA in water; mobile phase B: 0.1% FA in acetonitrile). The analytical column outlet was directly connected to the Nanospray Flex Ion Source. Up to top 10 precursors from the survey scan (350–2000 *m/z*, resolution 60,000) were selected for HCD fragmentation (target 50,000 charges; resolution 15,000, isolation window 2 *m/z*) with enabled dynamic exclusion (up to 45 s). Two independent LC-MS/MS analyses were done for on-beads digests. The analysis of the mass spectrometric RAW data files was carried out using the Proteome Discoverer software (Thermo Fisher Scientific; version 1.4) with in-house Mascot (Matrixscience; version 2.3.1) and Sequest search engines utilization. MS/MS ion searches were done against the UniProtKB protein database for mouse downloaded from ftp://ftp.uniprot.org/pub/database/uniprot/current_release/; version 20141001; 85,893 protein sequences

with additional sequences from cRAP database (www.thegpm.org/crap/). Mass tolerances for peptides and MS/MS fragments were 10 ppm and 0.05 Da, respectively. Oxidation of methionine and deamidation (N, Q) as optional modifications and two enzymes miss cleavages were set for all searches. Propionamidation of C as optional modification was set for database searches after in-gel digestion. Percolator was used for post-processing of Mascot search results. Peptides with false discovery rate (FDR; q-value) <1%, rank 1 and with at least six amino acids were considered. Proteins matching the same set of peptides were reported as protein groups. Protein groups/proteins were reported only if they had at least one unique peptide. Label-free quantification using protein area calculation in Proteome Discoverer was used (top 3 protein quantification). Two analyses of on-beads digests and analyses for all gel area within single gel line were searched as one dataset

**Wnt pathway activation readouts.** To analyze shift in DVL3 or ROR1 mobility upon treatment with conditioned medium, MEF cells were seeded on 24-well plates and grown to reach 70–80% confluency in DMEM medium supplemented with 10% FBS. Cells were treated overnight with recombinant human WNT3A (5036-WN-CF, R&D), WNT5A (645-WN-010, R&D), control-, Wnt3a- and Wnt5a-L-Cells-derived CM or CM obtained from TelChP- and HbChP- primary culture derived from either *Wnt5a*^WT or *Wnt5a*^cKO embryos. Twenty-four hour prior to the treatment cells was incubated with 1 μm/ml LGK974 inhibitor (974-02, StemRD). Used reagents are listed in the Supplementary Table 1. Used antibodies are listed in the Supplementary Table 3.

**Molecular sieve asay.** 100 ng of recombinant WNT5A were incubated with 0.6% CHAPS solution (C3023-5G, Sigma) or 50 ug of native LDL (LP3-5MG, Millipore) in 800 μl of serum-free DMEM for 4 h at 37 °C. Following incubation, sample was filtered through 150 kDa cut-off protein concentrator (PI89922, Thermo Scientific) at 20 °C, until dead-stop volume of ultrafiltrate was reached. This step was performed to separate high-molecular form of bound recombinant WNT5A from the unbound protein as described before for FGF protein[48]. The flow-through subjected to filtration through 3 kDa cut-off protein concentrator (PI88514S, Thermo Scientific) to concentrate the sample to the volume of ultrafiltrate obtained in the previous filtration step. Used reagents are listed in the Supplementary Table 1.

**Exosome purification.** Exosomes were purified by differential centrifugation as described previously[59]. In short, CM from E14.5 HbChP-derived primary culture was processed in series of centrifugation steps of 200 g, 1500 g, 6,000 g, and 14,000 g before pelleting exosomes at 100,000 g in SW55 swinging bucket (k factor —139) rotor for 2 h using Optima L-90 Xp Centrifuge (Beckman Coulter). Sucrose cushion step was included to increase the purity of exosomal fraction. The supernatant was discarded, and exosomes were resuspended in filtered PBS.

**Lipoproteins isolation and fractionation.** Lipoprotein isolation was performed using the protocol described previously[60]. In brief, lipoproteins from mouse serum and ChP primary culture were separated from soluble proteins using KBr discontinuous gradient. To remove excessive KBr, fractions were filtered using concentration columns (Millipore) and washed several times in PBS. Finally, the ultrafiltrate of lipoproteins fractions was resuspended in PBS before further applications. Used reagents are listed in the Supplementary Table 1.

**Delipidation.** Serum was mixed with suitable amount of LRA (13358-U, Sigma)[30] following slightly adapted manufacturer's instructions (80 g/L). Solution was mixed for 10 min, followed by two successive centrifugation steps at 8,000 g for 20 min to pellet LRA. In each step, the supernatant was carefully isolated to prevent contamination with the LRA sedimented at the bottom of the tube. Finally, delipidated

supernatant corresponding to delipidated serum was aliquoted and stored for further use at −80 °C. Used reagents are listed in the Supplementary Table 1.

**Immunofluorescence and EdU staining**. For mouse embryo immunofluorescent analysis and in situ hybridization, *WT CD1* mice, *Wnt5a^KO* or *Wnt5a^cKO* mice were dissected and isolated embryos were transferred into ice-cold PBS, fixed in 4% paraformaldehyde (PFA) in PBS for several hours followed by several washes in ice-cold PBS and finally cryoprotected by sequential incubation in 15% and then 30% sucrose solutions. Embryos were next frozen in Tissue-Tek optimum cutting temperature (OCT) compound (25608-930, Sakura Fine-Tek) on dry ice. Serial 14 μm coronal sections were used for immunofluorescence analysis. Human fetal tissue for cryosectioning was immersion-fixed overnight in 4% PFA at 4 °C, then cryoprotected in sucrose before embedding in OCT compound and then 14 μm sections were cut using a Leica cryostat. Human fetal tissue samples were processed using an identical immunofluorescence protocol as indicated for the mouse samples.

For immunofluorescent analysis, all the sections underwent antigen retrieval by direct boiling for 10 min at 550 W in the microwave using antigen retrieval solution (DAKO). Sections were washed in PBT (PBS with 0.5% Tween-20) and blocked in PBTA (PBS, 5% donkey serum, 0.3% Triton X-100, 1% BSA). Samples were incubated overnight at 4 °C with primary antibodies diluted in PBTA. Following washes in PBT, samples were incubated with corresponding Alexa Fluor secondary antibodies (Invitrogen) for 1 h at RT, followed by 5 min incubation at RT with DAPI (1:5000). Finally, samples were mounted in DAKO mounting solution (DAKO).

EdU (Life Technologies) was injected 72 h before the embryos were harvested at a concentration of 65 mg/g. Cells with incorporated EdU were visualized using a Click-iT EdU Alexa Fluor 555 Imaging Kit (C10338, Life Technologies).

For immunocytofluorescent analysis of ChP primary culture, cells grown on laminin coated cover slips were first washed several times in ice-cold PBS, followed by fixation for 15 min in ice-cold 4% PFA. Later, cells were washed several times in PBT, blocked with PBTA for 30 min and incubated overnight at 4 °C with primary antibodies. Following repeated washing in PBT cells have been incubated for 1 h at RT with appropriate secondary antibodies, DAPI for 5 min and mounted in DAKO mounting medium (DAKO). Used reagents are listed in the Supplementary Table 1. Used antibodies are listed in the Supplementary Table 3.

Processing of immunofluorescent images and quantitative assessment of signal overlap for WNT5A puncta was performed using vesicles colocalization function as a part of microscope image analysis software Imaris© (Bitplane). Parameters used for image processing were as follows: 0.4 μm cut-off for analyzed particles size, 0.1 distance for the signal overlap analysis using vesicles-colocalization function. Quantification of KI67+ and EdU staining was performed using ImageJ software.

**Transmission electron microscopy analysis**. For negative contrasting exosomes were adsorbed at activated formwar membrane coated with carbon EM grids (Pyser–SGI Limited), stained with 2% solution of ammonium molybdate, and visualized using transmission electron microscopy (TEM) Morgagni 268D (FEI) equipped with Mega ViewIII (Soft Imaging System) at 70 kV.

**In situ analysis**. In situ analysis of the gene expression was done on 14 μm cryosections of embryos at various stages of embryonic development isolated from CD1 mice. After isolation, embryos were immediately transferred and kept in fresh 4% PFA for 2 h, washed briefly in ice-cold PBS, incubated for 6 h in 30% sucrose solution at 4 °C, and frozen at −80 °C. Transcripts were detected using the RNAscope 2.0 assay for fresh frozen tissue (Advanced Cell Diagnostics). Staining was performed using the RNAscope Fluorescent Multiplex Reagent Kit (320850, Advanced Cell Diagnostics). Used reagents are listed in the Supplementary Table 1.

Indicated in situ images were adopted from Allen Institute for Brain Science: Allen Developing Mouse Brain Atlas[61] (Available from: http://developingmouse.brain-map.org) or from eurexpress.org[62] (Available from: http://www.eurexpress.org/ee/).

**Real-time qPCR**. RNA was isolated from 3 different litters of *WT CD1* embryos collected at indicated developmental stages. Samples were treated with DNase (M6101, Promega) to prevent contamination with genomic DNA. The specificity of primers was determined by BLAST run of the primer sequences. The sequences of primers are displayed in the Supplementary Table 4 and their annealing temperature is 57 °C for all used primers

qPCR reactions were performed once for every gene/sample in triplicate. PCR was done according to the manufacturer's protocol (Lightcycler 480 SYBR Green 1 Master Mix,Roche). The following thermo cycling program parameters were used for qPCR analysis: Incubation step at 95 °C for 5 min, then 45 cycles 95 °C for 10 s, 57 °C for 10 s, and 72 °C for 10 s. qPCR analysis was carried out using LightCycler© 480 Instrument II (Roche).

ΔCp values were calculated in every sample for each gene of interest with *β-Actin* as the reporter gene. Relative change of expression level for analyzed gene (ΔCp) was performed by subtraction of the gene expression levels in TelChP or HbChP from the gene expression level of housekeeping gene (*β-actin*). Next ratio

of the gene expression level between *β-actin* and gene of interest in either HbChP or TelChP was calculated using following formula: $2^{-\Delta C_P}$

**Statistics**. Gene expression data—Replicates are independent experiments. Data in Fig. 1c, Fig. 3a, Supplementary Fig. 5a and 5b are expressed as columns showing the mean with standard deviation (s.d.). Significance measured with paired two-tailed Student's *t*-test with unequal sample variance. Biological replicates per condition are indicated in the corresponding graphs.

Confocal images—Images used for quantitative analyses reported in Fig. 5c and Fig. 5d are representative images chosen from the results obtained in three consecutive sections obtained from three individual embryos processed in three separate experiments. Confocal images reported in Fig. 7e, g are representative images chosen from results obtained in three individual embryos. Confocal images reported in Supplementary Fig. 11a representative images chosen from the results obtained from six individual embryos. Confocal images reported in Supplementary Fig. 11c, e, g and h are representative images chosen from the results obtained in three individual embryos. Data in Fig. 5e, f, Fig. 7h-k and Supplementary Fig. 11b, d and f are expressed as columns showing the mean with s.d. Significance measured with unpaired or paired two-tailed Student's *t*-test with unequal sample variance. Except for Fig. 5f, biological replicates or sections used per condition are indicated in the corresponding graphs.

Western blot images—Image used for quantitative analysis reported in Fig. 6l is a representative image chosen from the results obtained in three separate experiments. Significance measured with unpaired two-tailed Student's *t*-test with unequal sample variance. Biological replicates per condition are indicated in the corresponding graph.

All the displayed immunostaining images and western blots are representative of at least three independent experiments.

**Reporting summary**. Further information on experimental design is available in the Nature Research Reporting Summary linked to this article.

## Data availability
The authors declare that all data supporting the findings of this study are available within this published article and its Supplementary Information files and from the corresponding author upon reasonable request. In situ hybridization data used in Fig. 1a, b and Supplementary Figs 1a, b and 3c are available from the Allen Developing Mouse Brain Atlas (www.alleninstitute.org). In situ hybridization data used in Supplementary Figs 2a–d and 10a–b are available from the Eurexpress atlas (www.eurexpress.org). The mass spectrometry proteomics data have been deposited at the ProteomeXchange Consortium via the PRIDE partner repository[63] with the dataset identifier PXD011918. Figures are deposited in figshare depository with the identifier [https://doi.org/10.6084/m9.figshare.7588481.v2]. A reporting summary for this Article is available as a Supplementary Information file. The source data underlying Figs. 1c, 3a, 5e–f, 6m, 7h–k and Supplementary Figs 5a–b, 7b; and 11b, d and f are provided as Source Data File.

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

## Acknowledgements

We thank Nadia Wänn for maintenance of mice colonies; the members of Bryja and Arenas lab for their help and suggestions; Martin Häring for help with in situ analysis; Johnny Söderlund and Alessandra Nanni for their technical and secretarial assistance; and the BIC imaging facility at Karolinska Institutet for technical support. We thank MEYS CR for support to the following core facilities: Proteomics (CIISB research infrastructure project LM2015043), cellular imaging at CEITEC institution at Masaryk University (LM2015062 Czech-BioImaging) Czech Centre for Phenogenomics (LM2015040), Higher quality and capacity of transgenic model breeding (by MEYS and ERDF, OP RDI CZ.1.05/2.1.00/19.0395), Czech Centre for Phenogenomics: developing

towards translation research (by MEYS and ESIF, OP RDE CZ.02.1.01/0.0/0.0/16_013/0001789). The collaboration between Masaryk University and Karolinska Institutet (KI-MU program), was co-financed by the European Social Fund and the state budget of the Czech Republic (CZ.1.07/2.3.00/20.0180). Funding to the VB lab was obtained from Neuron Fund for Support of Science (23/2016), and Czech Science Foundation (GA17-16680S). Work in the EA lab was supported by the Swedish Research Council (VR projects: DBRM, 2011-3116, 2011-3318 and 2016-01526), Swedish Foundation for Strategic Research (SRL program and SLA SB16-0065), European Commission (NeuroStemcellRepair), Karolinska Institutet (SFO Strat Regen, Senior grant 2018), Hjärn-fonden (FO2015:0202 and FO2017-0059) and Cancerfonden (CAN 2016/572). Research in the J.C.V. lab was supported by Karolinska Institutet Foundations. K.K. was supported by Masaryk University (MUNI/E/0965/2016). D.P. and Z.Z. were supported by the CEITEC 2020 (LQ1601) project from MEYS CR.

## Author contributions

K.K. designed and performed most of the experiments, analyzed data, prepared all figures, and wrote the manuscript; D.G. helped with the *Wnt5a* mouse strain and designed schematic depictions; J.P., M.P., and R.S. produced *Wnt5a*$^{cKO}$ strain; P.K. performed immunostaining analysis; A.S., T.R., and J.H. contributed to the IP experiments; R.L.G. and R.A.B. contributed the human fetal samples; D.P. and Z.Z. performed the MS analysis; F.L.M. and A.G.C. collected the mouse embryonic fluid; V.B., E.A., and J.C.V. designed experiments, supervised the work, analyzed data, and wrote the manuscript.

## Additional information

**Competing interests:** The authors declare no competing interests.

