## [Peer Review File · Nature Communications]

Reviewers' Comments:

Reviewer #1:

Remarks to the Author:

In this manuscript, the authors provide evidence that Wnt5a is expressed in the developing mouse and human hindbrain choroid plexus (HbChP). In the developing mouse embryo, Wnt5a is transported via cerebrospinal fluid (CSF) to dorsal hindbrain progenitor cells lining the ventricle where it inhibits proliferation of neural progenitors. They furthermore provide evidence that lipoproteins, but not exosomes, from the ChP transports Wnt5a in the CSF. This particular mechanism of WNT transport over long distances is very topical and important to the field. Overall, these experiments are quite convincing and the conclusions are largely supported by the data (some points of criticism are listed below).

A central question that arises from these experiments is the following: what is Wnt5a's function in the developing brain? The authors provide some insight by showing that Wnt5a inhibits proliferation of neural progenitors in the neuroepithelium in direct contact with the CSF. This data is quite compelling, but unfortunately the question remains on what Wnt5a does during brain development. Presumably an increase in proliferation due to lack of Wnt5a should manifest itself later in development or in the adult mouse. To my knowledge, a clear neurodevelopmental defect has not been identified in Wnt5a mutant mice, which casts some doubt on the importance of Wnt5a in this tissue. This is a difficult question to answer and, in my opinion, beyond the scope of this paper (presumably future studies from this lab will focus on Wnt5a's role in patterning the developing brain). For the purposes of this particular paper, I would ask the authors to dedicate a paragraph in the discussion on Wnt5a's potential role in neurodevelopment.

Additional comments:

The experiments showing that Wnt5a is secreted from HbChP (Figure 4) are convincing. However, experiments showing that Wnt5a is active are entirely correlative. The authors could strengthen their conclusion by showing that such activity is absent in cultured HbChP cells derived from Wnt5a mutant cells. Material for such an experiment should be readily available since they used it to show that Wnt5a is on lipoprotein complexes from WT but not Wnt5a mutants. Alternatively, the Wnt5a antibody used in these studies appears to be quite specific; have the authors tested whether it blocks this Wnt5a activity on Dvl3 phosphorylation? Additionally, treatment with a Porcn inhibitor would build confidence that the Dvl3 mobility shift is mediated by Wnt protein, potentially by Wnt5a.

Supplementary Figure 1a: it appears that Wnt5b is expressed in a similar manner to Wnt5a. This raises the potential for redundancy, especially given the high degree of similarity between these two genes. This could in part explain the weak phenotypic effects in the developing brain observed in Wnt5a knockout mice. Expanding their studies to include Wnt5b would be too much additional work and beyond the scope of this paper, however, at minimum the authors should mention Wnt5b's expression in the HbChP and discuss the possibility for redundancy.

The data that Wnt5a is not associated with exosomes are quite clear. However, the association with lipoprotein complexes is less convincing: it appears that only a fraction of Wnt5a is associated with APOA, B, E or J. A gel filtration analysis may provide insight into whether Wnt5a is in a small or large complex. More importantly, is the Wnt5a associated with lipoproteins active? Or is the free Wnt5a active? Or both? Are Wnt5a-lipoprotein complexes as isolated in Figure 6a active in the Dvl3 phosphorylation assay?

Experiments shown in Figure 6c-e do not strengthen the argument that lipoprotein particles are necessary for Wnt5a transport. Wnt proteins are highly hydrophobic and removal of lipids from culture media will diminish their solubility, and hence less Wnt protein will be detected upon lipid removal. "Rescuing" Wnt5a solubility by addition of normal serum or HDL or LDL simply adds back sufficient lipid such that the hydrophobic Wnt protein is once again soluble. These results are

entirely consistent with prior findings and do not establish that Wnt5a is in fact associated with lipoprotein particles.

Minor comments:

Figure 1a and 1b (and throughout the manuscript): For those unfamiliar with brain anatomy, it is difficult to identify the structures the authors refer to. I suggest a cartoon pictograph, or some judicious use of boxes and labels.

Figure 1c: I find the authors' choice of display here confusing. They argue that Wnt5a is not expressed in the TelChP, yet they set expression here at 1. Perhaps a mRNA relative to loading control (ΔCt) would be more meaningful. What is the difference between the three white bars here?

Gene/protein designations are confusing. The authors appear to use italics to designate gene names but then switch to all upper-case and non-italicized letters for proteins. Generally, all upper-case is reserved for human genes and proteins.

There is no in-text reference to Figure 1e.

Supplementary Figure 4d: it is unclear what information this table provides.

Reviewer #2:

Remarks to the Author:

The manuscript by Kaiser and colleagues reports that a member of the Wnt family, Wnt5a, is transported into lipoprotein particles, which are released from the choroid plexus into the cerebral spinal fluid. These are intriguing and exciting results.

Wnt proteins are lipid modified secreted proteins, which carry a palmitate group rendering these proteins less soluble and might explain their limited diffusion rate in tissues. Studies showed that Wnts are released in exosomes or transported along cytonemes, very thin processes form primarily by actin filaments. However, it remains unclear whether Wnts are transported by other mechanisms.

Here the authors present strong evidence that Wnt5a is transported in lipid particles that are released from the choroid plexus from specific brain areas. These results could have very important implications for our current understanding on Wnts might have long-range effects in the brain but also in other tissues.

The manuscript is very well written and the experiments are clear. In my view, this study represents an important contribution to the Wnt field. Before the paper could be considered for publication in Nature Communications, the authors need to address important issues.

Specific comments:

1) Figure 3, the authors examined the levels of GPR177 in the CSF from choroid plexus from the developing hindbrain (HbChP) and telencephalon (TelChP) using western blots. They claimed that GPR177 is mainly present in the HbChP but there is substantial difference in the loading of the gels as indicated by the higher levels of beta-actin in samples from HbChP. The authors should present better gels.

2) In Figure 4d, the authors examined the level of Wnt activity from samples isolated from the HbChP and TelChP by measuring the phosphorylation of Dishevelled-3. The authors also need to show changes in Wnt activity using other markers such as level of Axin-2 expression, β -catenin levels or its phosphorylation.

- 3) The data presented clearly show the presence of Wnt5a in the lipoprotein fraction and not in exosomes. However, the immunoprecipitation experiments are not clear or clear (Figure 6b). Proper controls such as mock IPs are missing, which are crucial to rule out the possibility that the antibodies used are nonspecific.
- 4) In Figure 6b, the IP gels show the presence of two protein bands when the anti GFP antibody is used (lower right panel) even in samples that do not express ApoE-GFP. This result is intriguing and seems to imply that the IP was not very clean. The authors should explain these results and should also run proper negative controls for IP experiments (i.e. the use of mock IP using non specific IgGs).
- 5) The specificity of the GPR177 antibody should be demonstrated.
- 6) Supplementary figure 6, the images are fuzzy and out off focus. Better images should be provided.

Reviewer #3:

Remarks to the Author:

In this study, Karol Kaiser and coauthors address the relevant question of how lipid-modified proteins involved in long-range signaling are spread in developing tissues. As the authors summarize, there is evidence for lipoproteins, exosomes and carrier proteins as vehicles for such transfer processes in various contexts.

The main message of the study is that the Wnt5a protein is transported via lipoprotein particles in the cerebrospinal fluid of the developing central nervous system, to regulate neural progenitor proliferation. This is an attractive idea but the paper falls short in providing convincing experimental evidence for this. In principle, one should demonstrate that the long-range transport of biologically active Wnt in the central nervous system is affected upon perturbation of CSF lipoproteins, to make such a claim.

The data provided demonstrates the following: colocalization of Wnt5a with select apolipoproteins in choroid plexus epithelial cells in vitro and in developing mouse hindbrain sections in situ (but see point 2 below). In addition, there is biochemical evidence for Wnt5a cofractionation with FBS (fetal bovine serum) lipoproteins and co-precipitation of co-overexpressed Wnt5a and ApoE-GFP from human embryonic kidney cells that do not secrete lipoproteins.

The most problematic conceptual issue with the available data is that the authors confuse serum lipoproteins/apolipoproteins with cerebrospinal fluid lipoproteins/apolipoproteins. CSF is considered not to contain serum lipoprotein particles, such as LDL or VLDL. In fact, apolipoprotein B, the major apoprotein of these particles, is commonly used as an indication of blood contamination of CSF. Considering that Wnts are lipidated and known to associate with lipoprotein particles, it is not surprising that Wnt5a is able to cofractionate with FBS lipoproteins (Fig. 6). Since the primary cultures of choroid plexus epithelial cells were cultivated in FBS, the apolipoproteins studied may be of bovine serum origin.

Additional concerns:

1. Fig. 5a-b: if Wnt4a associates with lipoproteins, why are the ratios of the signals in the Western blot of INPUT vs. 100gSN not similar for Wnt5a and apolipoproteins A and J? Shouldn't there be a similar enrichment of both in 100gSN? The authors should provide electron micrographs of the purified exosome and lipoprotein fractions.
2. Fig. 5d,g, 7c: The antibodies against apolipoproteins are not properly validated. Control without primary antibody (i.e. secondary ab only) is not sufficient because it does not exclude cross-reactivities of the primary antibody. Similar stringent controls as for the validation of Wnt5a antibody should be used. The image quantification in 5e is apparently only from a single embryo, which is too small a data set.

3. Fig. 6c: Preparation of delipidated serum (i.e. depletion of lipoproteins) should be performed by sequential ultracentrifugation (e.g. Goldstein JL et al. *Methods Enzymol.* 1983;98:241–260).

4. Fig. 3d. What do the authors mean by “high degree of correlation between Wnt5a and Gpr177 levels” in the images? There is not a high degree of colocalization and protein levels are not easy to judge from fluorescence micrographs.

Reviewer #4:

Remarks to the Author:

In the manuscript entitled: “Wnt5a is transported via lipoprotein particles in the cerebrospinal fluid and regulates neural progenitor proliferation” by Kaiser K. et al. the authors explored Wnt5a in the brain. They described that the choroid plexus in the developing hindbrain, but not in the telencephalon, produces Wnt5a in both mouse and human. They found Wnt5a associated with lipoprotein particles in cerebrospinal fluid and concluded that Wnt5a regulates proliferation of hindbrain neural progenitor cells. The observation is interesting but there is a severe lack of mechanism in that paper.

Specific comments:

Wnt5a signals through canonical and/or non-canonical pathways. In the canonical pathway, the signal is transduced to β -catenin, which enters the nucleus and forms a complex with TCF to activate transcription of Wnt target genes. Here, authors do not demonstrated which of the pathway is preferentially, or exclusively activated by Wnt5a. Nuclear β -catenin translocation and activation of Wnt target genes should be tested. The fact that in dorsal hindbrain cells express some Wnt/PCP signaling components is not sufficient to understand the mechanism. Moreover, if Wnt5a binds to Fzd3 and/or Frz10 to signal in the hindbrain, this needs to be determinate.

There is also a serious lack of mechanism regarding how Wnt5a signals in hindbrain and not in the telencephalon. For instance, authors suggest that Wnt5a is found associated with HDL and LDL fractions, however is this physiologically relevant? If Wnt5a regulates proliferation of hindbrain neural progenitor cells, are there any HDL and/or LDL receptors in the hindbrain that maybe not expressed in the telencephalon? Authors also pretend that Wnt5a co-immunoprecipitate with apoE, so it should be associated with apoE receptors such as LRP1 or other member of the LDL gene family that bind apoE in the brain. ApoE is one of the main apolipoprotein in the brain, thus mice lacking apoE should be defective for Wnt5a transport in the cerebrospinal fluid, or maybe other lipoproteins might compensate the defect?

As currently understood, Wnts ligand behave as growth factors. They predominantly mediate signaling between neighboring cells. Wnt5a is also usually known to promote cell proliferation and the increase in Ki67 expression in Wnt5a KO cells is weak and unusual. An inhibition of proliferation through Wnt5a is surprising and not in agreement with usual Wnt5a functions, and if this is the case, the authors should strengthen these data.

The term of “regulate” used through out the text is not precise enough and should be replaced by “up regulate” or “down regulate”. For example, in the abstract “... we found that Wnt5a regulates proliferation of hindbrain neural progenitor cells” does not tell the reader whether Wnt5a up regulates or down regulates proliferation. Here, it actually down regulates proliferation, which is obviously not in agreement with what is usually described for Wnt5a.

Reviewer 1

In this manuscript, the authors provide evidence that Wnt5a is expressed in the developing mouse and human hindbrain choroid plexus (HbChP). In the developing mouse embryo, Wnt5a is transported via cerebrospinal fluid (CSF) to dorsal hindbrain progenitor cells lining the ventricle where it inhibits proliferation of neural progenitors. They furthermore provide evidence that lipoproteins, but not exosomes, from the ChP transports Wnt5a in the CSF. This particular mechanism of WNT transport over long distances is very topical and important to the field. Overall, these experiments are quite convincing and the conclusions are largely supported by the data (some points of criticism are listed below).

A central question that arises from these experiments is the following: what is Wnt5a's function in the developing brain? The authors provide some insight by showing that Wnt5a inhibits proliferation of neural progenitors in the neuroepithelium in direct contact with the CSF. This data is quite compelling, but unfortunately the question remains on what Wnt5a does during brain development. Presumably an increase in proliferation due to lack of Wnt5a should manifest itself later in development or in the adult mouse. To my knowledge, a clear neurodevelopmental defect has not been identified in Wnt5a mutant mice, which casts some doubt on the importance of Wnt5a in this tissue. This is a difficult question to answer and, in my opinion, beyond the scope of this paper (presumably future studies from this lab will focus on Wnt5a's role in patterning the developing brain). For the purposes of this particular paper, I would ask the authors to dedicate a paragraph in the discussion on Wnt5a's potential role in neurodevelopment.

Reply: We thank Reviewer 1 for the positive evaluation of our manuscript. There is abundant evidence clearly showing that Wnt5a is important for proper development of CNS^{1,2,3,4}. Multiple publications describe the function of Wnt5a in the midbrain-hindbrain

region, some from our own group. In line with the reviewer suggestion we have incorporated a paragraph describing the role of Wnt5a in neural development with these references into the manuscript. Furthermore, in the current version of the manuscript we are also presenting clear evidence for a phenotypical effect of Wnt5a ablation in the development of cerebellum, using 2 different mouse models providing complementary data, thus supporting the role for the WNT5A in the embryonic development of CNS (Fig. 7f-k).

Additional comments:

The experiments showing that Wnt5a is secreted from HbChP (Figure 4) are convincing. However, experiments showing that Wnt5a is active are entirely correlative. The authors could strengthen their conclusion by showing that such activity is absent in cultured HbChP cells derived from Wnt5a mutant cells. Material for such an experiment should be readily available since they used it to show that Wnt5a is on lipoprotein complexes from WT but not Wnt5a mutants. Alternatively, the Wnt5a antibody used in these studies appears to be quite specific; have the authors tested whether it blocks this Wnt5a activity on Dvl3 phosphorylation? Additionally, treatment with a Porcn inhibitor would build confidence that the Dvl3 mobility shift is mediated by Wnt protein, potentially by Wnt5a.

Reply: In order to address this question, we have generated conditional knock model for ablation of WNT5A specifically in the choroid plexus (see Fig. 4 e-j). Analysis of conditioned medium from primary cultures generated from these mice clearly showed that the activity is associated with Wnt-5a (and not other Wnt in the primary culture; Fig. 4k) and with Wnt-5a from the ChP (and not other part of the embryonic brain).

Supplementary Figure 1a: it appears that Wnt5b is expressed in a similar manner to Wnt5a. This raises the potential for redundancy, especially given the high degree of similarity between these two genes. This could in part explain the weak phenotypic effects in the developing brain observed in Wnt5a knockout mice. Expanding their studies to include Wnt5b would be too much additional work and beyond the scope of this paper, however, at minimum the authors should mention Wnt5b's expression in the HbChP and discuss the possibility for redundancy.

Reply: In order to address the possibility of redundancy between Wnt5a and Wnt5b in the hindbrain choroid plexus we have analyzed the expression of Wnt5b in this tissue presented in the Supplementary Fig. 2. These data show negligible expression of Wnt5b in the ChP (Supplementary Fig. 2a'') and do not support the possibility of redundancy. Nevertheless, the issue of redundancy is discussed in the new version of the manuscript (on p.4) and accompanied with abovementioned Supplementary figure (Supplementary Fig.2).

The data that Wnt5a is not associated with exosomes are quite clear. However, the association with lipoprotein complexes is less convincing: it appears that only a fraction of Wnt5a is associated with APOA, B, E or J. A gel filtration analysis may provide insight into whether Wnt5a is in a small or large complex. More importantly, is the Wnt5a associated with lipoproteins active? Or is the free Wnt5a active? Or both? Are Wnt5a-lipoprotein complexes as isolated in Figure 6a active in the Dvl3 phosphorylation assay?

Reply: We thank reviewer for this comments that we addressed in the novel Fig. 6i-l . In this experiment we have produced Wnt5a/LDL complexes, separated free Wnt-5a and Wnt-5a-bound to LDL and tested the activity using Dvl3-phosphorylation assay. This experiment, in line with other indirect observations, demonstrates that WNT5A incorporated into LDL particles still shows activity (Fig. 6 i-l).

Experiments shown in Figure 6c-e do not strengthen the argument that lipoprotein particles are necessary for Wnt5a transport. Wnt proteins are highly hydrophobic and removal of lipids from culture media will diminish their solubility, and hence less Wnt protein will be detected upon lipid removal. “Rescuing” Wnt5a solubility by addition of normal serum or HDL or LDL simply adds back sufficient lipid such that the hydrophobic Wnt protein is once again soluble. These results are entirely consistent with prior findings and do not establish that Wnt5a is in fact associated with lipoprotein particles.

Reply: We thank reviewer for pointing this possibility – we have toned down our conclusions but still kept the data in the manuscript because they, in our opinion, provide in combination with other datasets mutually supporting lines of evidence for our claims.

Minor comments:

Figure 1a and 1b (and throughout the manuscript): For those unfamiliar with brain anatomy, it is difficult to identify the structures the authors refer to. I suggest a cartoon pictograph, or some judicious use of boxes and labels.

Reply: In the revised version we included cartoon scheme of anatomic localization of the telencephalic and hindbrain choroid plexus within the developing CNS as requested (Fig.1a, Supplementary Fig.1a).

Figure 1c: I find the authors’ choice of display here confusing. They argue that Wnt5a is not expressed in the TelChP, yet they set expression here at 1. Perhaps a mRNA relative to loading control (deltaCt) would be more meaningful. What is the difference between the three white bars here?

Reply: The presentation and the legend in Fig.1c have been improved as suggested.

Gene/protein designations are confusing. The authors appear to use italics to designate gene names but then switch to all upper-case and non-italicized letters for proteins. Generally, all upper-case is reserved for human genes and proteins.

Reply: In the revision we have adopted official Uniprot nomenclature for the mouse protein designation in order to avoid confusion regarding the labeling.

There is no in-text reference to Figure 1e.

Reply: The reference to Fig. 1f (former Fig. 1e) has been added to the revised text.

Supplementary Figure 4d: it is unclear what information this table provides.

Reply: We agree with the referee and have removed the table from Supplementary Fig. 7d (former 4d).

Reviewer2:

The manuscript by Kaiser and colleagues reports that a member of the Wnt family, Wnt5a, is transported into lipoprotein particles, which are released from the choroid plexus into the cerebral spinal fluid. These are intriguing and exciting results.

Wnt proteins are lipid modified secreted proteins, which carry a palmitate group rendering these proteins less soluble and might explain their limited diffusion rate in tissues. Studies showed that Wnts are released in exosomes or transported along cytonemes, very thin processes form primarily by actin filaments. However, it remains unclear whether Wnts are transported by other mechanisms.

Here the authors present strong evidence that Wnt5a is transported in lipid particles that are released from the choroid plexus from specific brain areas. These results could have very important implications for our current understanding on Wnts might have long-range effects in the brain but also in other tissues.

The manuscript is very well written and the experiments are clear. In my view, this study represents an important contribution to the Wnt field. Before the paper could be considered for publication in Nature Communications, the authors need to address important issues.

Reply: We thank reviewer 2 for the positive evaluation of our manuscript and we address the individual points as specified below.

Specific comments:

1) Figure 3, the authors examined the levels of GPR177 in the CSF from choroid plexus from the developing hindbrain (HbChP) and telencephalon (TelChP) using western blots. They claimed that GPR177 is mainly present in the HbChP but there is substantial difference in the loading of the gels as indicated by the higher levels of beta-actin in samples from HbChP. The authors should present better gels.

Reply: In order to address this point relating to the Western blot from Fig.3c (former Fig. 3b) has been replaced by western blot with better loading control.

2) In Figure 4d, the authors examined the level of Wnt activity from samples isolated from the HbChP and TelChP by measuring the phosphorylation of Dishevelled-3. The authors also need to show changes in Wnt activity using other markers such as level of Axin-2 expression, β -catenin levels or its phosphorylation.

Reply: We thank reviewer for this important point regarding the activation of the Wnt/ β -catenin pathway. During the revision we have analyzed activation of multiple readouts of canonical Wnt pathway by conditioned medium from ChP primary cells. Novel data in Fig.4k and (Supplementary Figure 5) clearly suggest that canonical Wnt pathway is not activated – as shown at mRNA level of β -catenin targets (TCF1 and Axin2; Supplementary Fig. 5a,b) as well as at the protein level (destabilization of Axin1 and dephosphorylation of β -catenin detected by active- β -catenin antibody; Fig. 4k). Wnt3a has been used as a positive control.

3) The data presented clearly show the presence of Wnt5a in the lipoprotein fraction and not in exosomes. However, the immunoprecipitation experiments are not clear or clear (Figure 6b). Proper controls such as mock IPs are missing, which are crucial to rule out the possibility that the antibodies used are nonspecific.

4) In Figure 6b, the IP gels show the presence of two protein bands when the anti GFP antibody is used (lower right panel) even in samples that do not express ApoE-GFP. This result is intriguing and seems to imply that the IP was not very clean. The authors should explain these results and should also run proper negative controls for IP experiments (i.e. the use of mock IP using nonspecific IgGs).

Reply to point 3 & 4: In order to clarify protein-protein interaction of apolipoproteins and Wnt5a we have redone all the co-IPs with better expression constructs as well as better control (non-specific IgG) to the immunoprecipitation experiments, which are now included

in the revised version of the figure (Fig. 6 b-e). This dataset clearly demonstrates interaction of Wnt-5a with ApoJ and ApoE.

5) The specificity of the GPR177 antibody should be demonstrated.

Reply: During the work on the revision the specificity of the GPR177 antibody has been validated using various tagged GPR177 (referred to in the new re-submitted version of the manuscript as WIs) constructs (Supplementary Fig. 4a). Moreover, we also corroborated our results from immunostaining with in situ analysis that is now available in the new version of the manuscript.

6) Supplementary figure 6, the images are fuzzy and out of focus. Better images should be provided.

Reply: Better quality in situ pictures from online database presented in the Supplementary Fig.10 (former Supplementary Fig.6).

Reviewer 3

In this study, Karol Kaiser and coauthors address the relevant question of how lipid-modified proteins involved in long-range signaling are spread in developing tissues. As the authors summarize, there is evidence for lipoproteins, exosomes and carrier proteins as vehicles for such transfer processes in various contexts.

The main message of the study is that the Wnt5a protein is transported via lipoprotein particles in the cerebrospinal fluid of the developing central nervous system, to regulate neural progenitor proliferation. This is an attractive idea but the paper falls short in providing convincing experimental evidence for this. In principle, one should demonstrate that the long-range transport of biologically active Wnt in the central nervous system is affected upon perturbation of CSF lipoproteins, to make such a claim.

The data provided demonstrates the following: colocalization of Wnt5a with select apolipoproteins in choroid plexus epithelial cells in vitro and in developing mouse hindbrain sections in situ (but see point 2 below). In addition, there is biochemical evidence for Wnt5a cofractionation with FBS (fetal bovine serum) lipoproteins and coprecipitation of co-overexpressed Wnt5a and ApoE-GFP from human embryonic kidney cells that do not secrete lipoproteins.

The most problematic conceptual issue with the available data is that the authors confuse serum lipoproteins/apolipoproteins with cerebrospinal fluid lipoproteins/apolipoproteins. CSF is considered not to contain serum lipoprotein particles, such as LDL or VLDL. In fact,

apolipoprotein B, the major apoprotein of these particles, is commonly used as an indication of blood contamination of CSF. Considering that Wnts are lipidated and known to associate with lipoprotein particles, it is not surprising that Wnt5a is able to cofractionate with FBS lipoproteins (Fig. 6). Since the primary cultures of choroid plexus epithelial cells were cultivated in FBS, the apolipoproteins studied may be of bovine serum origin.

Reply: We agree with the reviewer that ApoB is actually used as a marker of contamination of CSF with plasma. This evidence however, comes from studies in adults, where the barrier-like properties of choroid plexus and subsequent access of different molecular complexes to enter CSF via this site are different from the situation during the embryogenesis⁵. We refer to several publications (Zappaterra et al., 2007 – ref.42., Parada et al., 2008-ref.43 in the manuscript) that provide direct support to the presence of ApoB LDL particles in the embryonic CSF. We also provide a separate “for review only” file (p1) with representative images of immunostaining for ApoB demonstrating presence of punctate structures along the apical region of ventricular zone of neuroepithelium. In addition, we add in this file extra literature references that also identified ApoB presence in the embryonic CSF across several species. Also, the reviewer asked for the validation of the HbChP origin of the apolipoproteins, e.g. ApoE and ApoJ, detected in the fraction shown in Fig. 6a. In the revision we demonstrate specificity of the signal and its origin from HbChP primary cultures in the newly added Supplementary Fig. 8h, i.

Additional concerns:

1. Fig. 5a-b: if Wnt4a associates with lipoproteins, why are the ratios of the signals in the Western blot of INPUT vs. 100gSN not similar for Wnt5a and apolipoproteins A and J? Shouldn't there be a similar enrichment of both in 100gSN? The authors should provide electron micrographs of the purified exosome and lipoprotein fractions.

Reply: The signal intensity of WNT5A in Input and 100gSN is not consistently different as it appears in Fig. 5b. We provide results of an independent experiment (for review only file – p2) that clearly shows that levels of Wnt5a in these two fractions does not differ. We also provide electron micrograph images of exosomes (Supplementary Figure 6) to further demonstrate that the exosomal fraction obtained by our ultracentrifugation-based protocol indeed contains exosomes.

2. Fig. 5d,g, 7c: The antibodies against apolipoproteins are not properly validated. Control without primary antibody (i.e. secondary ab only) is not sufficient because it does not exclude cross-reactivities of the primary antibody. Similar stringent controls as for the

validation of Wnt5a antibody should be used. The image quantification in 5e is apparently only from a single embryo, which is too small a data set.

Reply: During our work on the revision we have overexpressed individual apolipoproteins and, on these samples, validated the specificity of antibodies to show that they do not mutually cross-react, do not cross-react with human apolipoproteins and do recognize only one particular apolipoprotein (Supplementary Fig. 8 a-g). We also provide IgG controls for immunostaining analysis as requested (Fig. 5d; Supplementary Fig. 7c). In addition, we expanded the original dataset used for the co-localization analysis (Fig.5e) to include 3 embryos instead of 1.

3. Fig. 6c: Preparation of delipidated serum (i.e. depletion of lipoproteins) should be performed by sequential ultracentrifugation (e.g. Goldstein JL et al. Methods Enzymol. 1983;98:241–260).

Reply: During the revision we have attempted to use the alternative delipidation protocol suggested by the reviewer. Unfortunately, we did not manage to detect any difference in the protein abundance for various apolipoproteins between different conditions using this approach which suggests that the protocol did not work in our hands. Nevertheless, we have implemented another approach for delipidation (this protocol is based on aqueous and organic phase separation using mixture of butanol and di-isopropyl ether , described in more detail in ref.22 in the manuscript and based on the original publication⁶) and observed similar results as we presented in original manuscript – these data are provided as “for review only file p. 3”.

4. Fig. 3d. What do the authors mean by” high degree of correlation between Wnt5a and Gpr177 levels” in the images? There is not a high degree of colocalization and protein levels are not easy to judge from fluorescence micrographs.

Reply: We have corroborated our initial observation about the relationship between signal intensity for the Wnt5a and Wls in HbChP epithelium also at the level of transcription using in situ analysis (RNAScope) to demonstrate that the intensities of Wnt5a and Gpr177 indeed correlate at the single cell level (Fig. 3e,f).

Reviewer 4:

In the manuscript entitled: “Wnt5a is transported via lipoprotein particles in the cerebrospinal fluid and regulates neural progenitor proliferation” by Kaiser K. et al. the authors explored Wnt5a in the brain. They described that the choroid plexus in the developing hindbrain, but not in the telencephalon, produces Wnt5a in both mouse and

human. They found Wnt5a associated with lipoprotein particles in cerebrospinal fluid and concluded that Wnt5a regulates proliferation of hindbrain neural progenitor cells. The observation is interesting but there is a severe lack of mechanism in that paper.

Specific comments:

Wnt5a signals through canonical and/or non-canonical pathways. In the canonical pathway, the signal is transduced to β -catenin, which enters the nucleus and forms a complex with TCF to activate transcription of Wnt target genes. Here, authors do not demonstrate which of the pathway is preferentially, or exclusively activated by Wnt5a. Nuclear β -catenin translocation and activation of Wnt target genes should be tested. The fact that in dorsal hindbrain cells express some Wnt/PCP signaling components is not sufficient to understand the mechanism. Moreover, if Wnt5a binds to Fzd3 and/or Frz10 to signal in the hindbrain, this needs to be determinate.

Reply: In order to address in detail how Wnt5a produced by ChP signals we have performed more thorough analysis of the Wnt pathway activation by HbChP secreted Wnt5a with emphasis on the canonical signaling. The comprehensive analysis of the Wnt components activated by secreted WNT5A (Fig.4 e-k) and absence of induction of target genes for canonical Wnt pathway (Supplementary Fig. 5a,b) strongly indicates that Wnt5a in this context activates only non-canonical and not the canonical branch of Wnt pathway.

Unfortunately, within the review we were unable to address the complex question of Wnt5a interaction with Fzd receptors - Fzd3 or Fzd10 - in the hindbrain. However, we believe this is an important question which would require generation of additional knockout mice and their robust analysis and as such we think it is currently beyond the scope of this manuscript.

There is also a serious lack of mechanism regarding how Wnt5a signals in hindbrain and not in the telencephalon. For instance, authors suggest that Wnt5a is found associated with HDL and LDL fractions, however is this physiologically relevant? If Wnt5a regulates proliferation of hindbrain neural progenitor cells, are there any HDL and/or LDL receptors in the hindbrain that maybe not expressed in the telencephalon? Authors also pretend that Wnt5a co-immunoprecipitate with apoE, so it should be associated with apoE receptors such as LRP1 or other member of the LDL gene family that bind apoE in the brain. ApoE is one of the main apolipoprotein in the brain, thus mice lacking apoE should be defective for Wnt5a transport in the cerebrospinal fluid, or maybe other lipoproteins might compensate the defect?

Reply: In order to address this issue, we have checked the literature that characterizes ApoE KO mouse – no notable embryonic CNS phenotype similar to the ones described for the Wnt5a KO embryo has been reported. It is however possible, as the reviewer also

mentions, that this might be due to the functional redundancy of apolipoproteins in the CNS. In our opinion, a more detailed dissection of this question would require a large amount of additional work including extensive work in animal models and is thus beyond the scope of this revision. It is of interest, that the Lrp4 receptor, which is APOE dedicated binding receptor, is expressed predominantly in the hindbrain portion of the developing CNS (data not shown).

Considering currently accepted model for the dynamics of CSF circulation in the central nervous system with unidirectional rostrocaudal net flow of the fluid from the lateral ventricle via the 4th ventricle reaching the spinal cord, the effect of Wnt5a secreted into CSF by a hindbrain choroid plexus should be mostly restricted to the regions in direct proximity of the tissue or located more caudally as opposed to regions located for example in the telencephalon⁷.

As currently understood, Wnt ligands behave as growth factors. They predominantly mediate signaling between neighboring cells. Wnt5a is also usually known to promote cell proliferation and the increase in Ki67 expression in Wnt5a KO cells is weak and unusual. An inhibition of proliferation through Wnt5a is surprising and not in agreement with usual Wnt5a functions, and if this is the case, the authors should strengthen these data.

Reply: Wnt5a has been shown to both promote^{8,9} as well as inhibit¹⁰ proliferation and it seems that its activity is highly dependent on the context. Inspired by this comment, we have now undertaken additional detailed analysis of hindbrain regions in Wnt5a deficient knockout cells. We have introduced also a novel model of conditional, ChP-specific KO of Wnt5a and compared it to both full Wnt5a KO as well as wildtype. Intriguingly, we managed to show in the new version of the manuscript that the proliferation effect in the epithelium is coupled to significant reduction in the size of the developing cerebellum in both models of Wnt5a ablation, Wnt5a KO and Wnt5a cKO specific for plexus (Fig. 7 f-i). Closer mechanistic exploration of this results was not possible in the given time frame of the revision, however we strongly believe that this represents substantial expansion of the originally reported Wnt5a phenotype and provides strong evidence that Wnt5a produced in the HbChP affects proliferation in the hindbrain.

The term of “regulate” used throughout the text is not precise enough and should be replaced by “up regulate” or “down regulate”. For example, in the abstract “... we found that Wnt5a regulates proliferation of hindbrain neural progenitor cells” does not tell the reader whether Wnt5a up regulates or down regulates proliferation. Here, it actually down regulates proliferation, which is obviously not in agreement with what is usually described for Wnt5a

Reply. Too often use of the word “regulate” has been replaced with more specific terms as requested by the reviewer.

References

1. Andersson, E. R. *et al.* Wnt5a regulates ventral midbrain morphogenesis and the development of A9-A10 dopaminergic cells in vivo. *PLoS One* **3**, (2008).
2. Blakely, B. D. *et al.* Wnt5a regulates midbrain dopaminergic axon growth and guidance. *PLoS One* **6**, (2011).
3. Vivancos, V. *et al.* Wnt activity guides facial branchiomotor neuron migration, and involves the PCP pathway and JNK and ROCK kinases. *Neural Dev.* **4**, 7 (2009).
4. Subashini, C. *et al.* Wnt5a is a crucial regulator of neurogenesis during cerebellum development. *Sci. Rep.* **7**, 42523 (2017).
5. Dziegielewska, K. M., Ek, J., Habgood, M. D. & Saunders, N. R. Development of the choroid plexus. *Microsc. Res. Tech.* **52**, 5–20 (2001).
6. Cham, B. E. & Knowles, B. R. A solvent system for delipidation of plasma or serum without protein precipitation. *J. Lipid Res.* **17**, 176–181 (1976).
7. Sakka, L., Coll, G. & Chazal, J. Anatomy and physiology of cerebrospinal fluid. *Eur. Ann. Otorhinolaryngol. Head Neck Dis.* **128**, 309–316 (2011).
8. Bo, H., Gao, L., Chen, Y., Zhang, J. & Zhu, M. Upregulation of the expression of Wnt5a promotes the proliferation of pancreatic cancer cells in vitro and in a nude mouse model. *Mol. Med. Rep.* **13**, 1163–1171 (2016).
9. Shojima, K. *et al.* Wnt5a promotes cancer cell invasion and proliferation by receptor-mediated endocytosis-dependent and -independent mechanisms, respectively. *Sci. Rep.* **5**, 8042 (2015).
10. Zhang, J. *et al.* Wnt5a inhibits the proliferation and melanogenesis of melanocytes. *Int. J. Med. Sci.* **10**, 699–706 (2013).

Reviewers' Comments:

Reviewer #1:

Remarks to the Author:

Comments on Revised Manuscript:

The authors have added significant amounts of new data and thereby substantially improved their manuscript. New data includes addition of conditional Wnt5a mutant mice and mechanism of Wnt5a signaling. With this revision, the authors have adequately addressed my primary concerns. However, one point of confusion remains: data shown Figure 1c are now more confusing than in the original version, which I requested to be changed. As shown, it looks like Wnt5a transcript levels are higher in TeICHp than in HbChp, which is obviously inconsistent with in situ hybridization (Fig. 1 b,d) and immunoblot (Fig. 1e) data. Are transcript levels in each sample relative to a loading control, such as Gapdh, Rpl, Ef1a, etc? If yes, which control was used? This control should be indicated on the y-axis as a ratio (e.g. "Wnt5a/control") or in the figure legend. The same question/comment applies to WIs RT-qPCR data in Fig. 3a. This display of RT-qPCR data is highly unusual (I have never seen it presented this way) and the authors undermine their own claims by showing it this way.

Reviewer #2:

Remarks to the Author:

The authors have addressed all my criticisms and comments appropriately.

The authors have included a number of new experiments that in my view has further enhanced the impact of this paper.

I would strongly recommend publication in Nature Communications.

Reviewer #4:

Remarks to the Author:

Thank you for your modifications and additional experiments in the manuscript entitled: "Wnt5a is transported via lipoprotein particles in the cerebrospinal fluid and regulates neural progenitor proliferation". We have no additional comment except that we suggest that the physical interactions between Wnt5a and APOE and J should be strengthened by alternative techniques such as in vitro interactions using purified recombinant proteins, or mass spectrometry analysis.

Reply to the Referees:

Reviewer 1

The authors have added significant amounts of new data and thereby substantially improved their manuscript. New data includes addition of conditional Wnt5a mutant mice and mechanism of Wnt5a signaling. With this revision, the authors have adequately addressed my primary concerns. However, one point of confusion remains: data shown Figure 1c are now more confusing than in the original version, which I requested to be changed. As shown, it looks like Wnt5a transcript levels are higher in TelChP than in HbChP, which is obviously inconsistent with in situ hybridization (Fig.1 b,d) and immunoblot (Fig. 1e) data. Are transcript levels in each sample relative to a loading control, such as Gapdh, Rpl, Ef1a, etc? If yes, which control was used? This control should be indicated on the y-axis as a ratio (e.g. "Wnt5a/control") or in the figure legend. The same question/comment applies to Wls RT-qPCR data in Fig. 3a. This display of RT-qPCR data is highly unusual (I have never seen it presented this way) and the authors undermine their own claims by showing it this way.

Reply: We thank Reviewer 1 for the positive assessment of the improvements to the overall quality of the manuscript. Requested modifications have been implemented to all the qPCR data in question (Fig. 1c, Fig. 3a). We believe that current structure of the graph provides more comprehensible representation of the data, highlighting the key differences in expression patterns between choroid plexuses and satisfying all the key points made by the reviewer.

Reviewer 2:

The authors have addressed all my criticisms and comments appropriately.

The authors have included a number of new experiments that in my view has further enhanced the impact of this paper.

I would strongly recommend publication in Nature Communications.

Reply: We would like to thank Reviewer 2 for positive feedback on the manuscript.

Reviewer 4

The comments from your revisions to the points raised by Referee #3 from Referee #4 were: "I totally agree with comments from referee #3, and I think the best way to test whether Wnt5a binds to apoE or apoJ is to perform *in vitro* binding experiments using recombinant proteins (i.e.: recombinant Wnt5a, and recombinant apoE). This is what I suggested in my last comment.

In the reference 42 (Zappaterra M. D. et al., 2007) cited by authors, they performed a proteomic study of human embryonic cerebrospinal fluid and compare it to rat embryonic cerebrospinal fluid during development. They found that apoA1, apoAIV, apoE and apoB100 are present, but not apoJ nor Wnt5a. If Wnt5a binds to ApoE and ApoJ one could expect to find it also in this proteomic analysis"

--

Reviewer #4 (Remarks to the Author)

Thank you for your modifications and additional experiments in the manuscript entitled: "Wnt5a is transported via lipoprotein particles in the cerebrospinal fluid and regulates neural progenitor proliferation". We have no additional comment except that we suggest that the physical interactions between Wnt5a and APOE and J should be strengthened by alternative techniques such as *in vitro* interactions using purified recombinant proteins, or mass spectrometry analysis.

*Reply: We are grateful for practical suggestions which we agree would provide strong evidence for suggested link between WNT5A and APOE/APOJ proteins. Therefore, in the updated version of the manuscript we successfully demonstrated direct physical interaction under the *in vitro* condition between recombinant WNT5A and native form of both APOE and APOJ proteins. In our opinion this new data presented in Fig. 6 f/g fully addresses the questions raised by the reviewer.*

Additional concerns:

In the reference 42 (Zappaterra M. D. et al., 2007) cited by authors, they performed a proteomic study of human embryonic cerebrospinal fluid and compare it to rat embryonic cerebrospinal fluid during development. They found that apoA1, apoAIV, apoE and apoB100 are present, but not apoJ nor Wnt5a. If Wnt5a binds to ApoE and ApoJ one could expect to find it also in this proteomic analysis"

Reply: Regarding the highlighted absence of Wnt5a and apoJ in proteomic data reported in the reference 42 (Zappaterra M. D. et al., 2007; ref.43 in the updated version of the manuscript), we would like to firstly highlight the fact that alternative name for APOJ protein is Clusterin (this information has been now included into the updated version of

the manuscript). Indeed, Clusterin has been successfully identified to be present in the cerebrospinal fluid according to the reported mass spectrometry data presented in the article in line with the hypothesis presented in our manuscript.

Furthermore, we believe that the absence of *Wnt5a* can be, at least partially, explained as a result of limited sensitivity of the method which has been significantly improved since. This is highlighted by the substantial number of newly identified extracellular proteins that have been detected in the embryonic cerebrospinal fluid in recent years but were absent from the dataset presented in the original article (Zappaterra M. D. et al., 2007). These include for example *SHH*¹, *DKK2*² or *IGF2*³. Therefore, we believe that further advancement linked to sensitivity of the mass spectrometry approaches would allow to identify additional proteins secreted into embryonic cerebrospinal fluid comprising *WNT5A*.

References

1. Huang, X. et al. Transventricular delivery of Sonic hedgehog is essential to cerebellar ventricular zone development. *Proc. Natl. Acad. Sci.* **107**, 8422 LP-8427 (2010).
2. Johansson, P. A. et al. The transcription factor *Otx2* regulates choroid plexus development and function. *Development* **140**, 1055 LP-1066 (2013).
3. Lehtinen, M. K. et al. The cerebrospinal fluid provides a proliferative niche for neural progenitor cells. *Neuron* **69**, 893–905 (2011).

Reviewers' Comments:

Reviewer #4:

Remarks to the Author:

In their answers to reviewers authors mentioned interactions between recombinant Wnt5a and ApoE and J. These data are shown in the new figure 6f/g. There is a clear problem with these data. If apparently there is an interaction between ApoJ and Wnt5a using in vitro co IP experiments and recombinant proteins, there is a doubt on the interaction between ApoE and Wnt5a. Figure 6, panel g, shows that when Wnt5a is immunoprecipitated, Wnt5a is not found in the complex by western blot! This goes against author's conclusions and clearly shows that there are serious doubts about the specificity of the anti-Wnt5a used throughout the paper.

Reviewer 4

Reviewer #4 (Remarks to the Author):

“In their answers to reviewers authors mentioned interactions between recombinant Wnt5a and ApoE and J. These data are shown in the new figure 6f/g. There is a clear problem with these data. If apparently there is an interaction between ApoJ and Wnt5a using in vitro co IP experiments and recombinant proteins, there is a doubt on the interaction between ApoE and Wnt5a. Figure 6, panel g, shows that when Wnt5a is immunoprecipitated, Wnt5a is not found in the complex by western blot! This goes against author’s conclusions and clearly shows that there are serious doubts about the specificity of the anti-Wnt5a used throughout the paper”

We agree with reviewers comments regarding the confusing nature of the interaction between Wnt5a and ApoE as highlighted in the pull-down experiment shown in the original Fig.6g. We observed this discrepancy consistently in several repetitions of the experiments but cannot explain the underlying mechanism. To avoid confusion, we decided to remove the experimental data in question, e.g. Fig.6g, from the manuscript.

In addition, we would like to emphasize the fact that the Wnt5a antibody used for the immunoprecipitation experiments (e.g. AF645, R&D) is different from the Wnt5a antibody (MAB645, R&D) that has been validated extensively in the manuscript for all the other applications such as immunostaining and western blot. Therefore, the questionable result obtained with the Wnt5a antibody (AF645, R&D) in the pull-down experiments does not cast any doubts on the specificity and validity of the results

acquired in all the other experiments where the other Wnt5a antibody (MAB645, R&D) has been used.